# Tet1 and Tet2 maintain mesenchymal stem cell homeostasis via demethylation of the *P2rX7* promoter

Ruili Yang[1,2,3], Tingting Yu[1,2], Xiaoxing Kou[1,2], Xiang Gao[2,4], Chider Chen[2], Dawei Liu[1,2], Yanheng Zhou[1,3] & Songtao Shi[2,3]

Ten-eleven translocation (Tet) family-mediated DNA oxidation represents an epigenetic modification capable of converting 5-methylcytosine (5-mC) to 5-hydroxymethylcytosine (5-hmC), which regulates various biological processes. However, it is unknown whether Tet family affects mesenchymal stem cells (MSCs) or the skeletal system. Here we show that depletion of Tet1 and Tet2 results in impaired self-renewal and differentiation of bone marrow MSCs (BMMSCs) and a significant osteopenia phenotype. Tet1 and Tet2 deficiency reduces demethylation of the *P2rX7* promoter and downregulates exosome release, leading to intracellular accumulation of miR-297a-5p, miR-297b-5p, and miR-297c-5p. These miRNAs inhibit Runx2 signaling to impair BMMSC function. We show that overexpression of P2rX7 rescues the impaired BMMSCs and osteoporotic phenotype in *Tet1* and *Tet2* double knockout mice. These results indicate that Tet1 and Tet2 play a critical role in maintaining BMMSC and bone homeostasis through demethylation of *P2rX7* to control exosome and miRNA release. This Tet/P2rX7/Runx2 cascade may serve as a target for the development of novel therapies for osteopenia disorders.

[1] Department of Orthodontics, Peking University School & Hospital of Stomatology, #22 Zhongguancun South Avenue, Beijing 100081, China. [2] Department of Anatomy and Cell Biology, University of Pennsylvania, School of Dental Medicine, Philadelphia, PA 19104, USA. [3] Sino-US joint Research Center of Oral Tissue-derived Stem Cells, PKU Industrial Park, Building 10 First Floor, Beiqing Road, Changping District, Beijing 102200, China. [4] College of Stomatology and Chongqing Key Laboratory of Oral Diseases and Biomedical Sciences, Chongqing Medical University, Chongqing 401147, China. These authors contributed equally: Ruili Yang, Tingting Yu. Correspondence and requests for materials should be addressed to Y.Z. (email: yanhengzhou@vip.163.com) or to S.S. (email: songtaos@upenn.edu)

The ten-eleven translocation (Tet) family is a group of DNA demethylases capable of regulating various epigenetic responses. Tet proteins, including Tet1, Tet2, and Tet3, are able to convert 5-methylcytosine (5-mC) to 5-hydroxymethylcytosine (5-hmC) and its oxidative derivatives in Fe(II)- and alpha-ketoglutarate ($\alpha$-KG)-dependent oxidation reaction to promote DNA demethylation and gene transcription[1–4]. Previous studies showed that 5-hmC is abundant in both adult cells and embryonic stem cells (ESCs)[5–7]. Upon ESC differentiation, the expression levels of Tet1 and Tet2 are downregulated, suggesting that Tet1 and Tet2 may be associated with the maintenance of ESC pluripotency through regulation of lineage-specific genes[1]. It was reported that the expressions of Tet1 and Tet2 were regulated by Oct4/Sox2 complex, and the depletion of Tet1 impairs the self-renewal and differentiation of ESCs[5, 8]. In contrast to its role in maintaining ESC pluripotency, Tet proteins have different effects on adult stem cells. Hematopoietic stem cells (HSCs) from *Tet2*-mutant mice exhibit increased repopulating capacity, augmented HSC expansion, and impaired differentiation toward the myeloid lineage[9–11]. Moreover, Tet2 and 5-hmC levels are increased during smooth muscle cell differentiation[7]. Adult *Tet1*-mutant mice exhibit defective self-renewal of neural progenitor cells and impaired memory extinction[12]. These studies suggest that Tet proteins may possess unique functional roles in epigenetic regulation of stem cell function. Depletion of Tet1 resulted in a reduction of 5-hmC levels and impaired chondrogenic differentiation in a chondroprogenitor cell line[13]. Additionally, 75% of the newborn Tet1-depletion mice exhibit a smaller body size at birth but seemed to gain their body weight when growing to 6–9 weeks[14], suggesting a potential skeletal defect. However, the role of Tet proteins in skeletal development remains largely unknown.

Bone marrow mesenchymal stem cells (BMMSCs) are a population of non-hematopoietic multipotent stem cells with self-renewal and multipotent differentiation capacities. They play an essential role in maintaining bone/marrow homeostasis[15, 16]. BMMSCs can be regulated at both transcriptional and epigenetic levels in response to stimulation from various environmental elements[17–19]. BMMSC deficiency may contribute to bone degenerative phenotypes in osteopenia disease models[20, 21].

In this study, we show that Tet1 and Tet2 are required to maintain BMMSC and bone homeostasis. The depletion of Tet1 and Tet2 may lead to hypermethylation of the *P2rX7* promoter to block miR-297a-5p, miR-297b-5p, and miR-297C-5p release, leading to downregulation of Runx2 signaling and osteopenia phenotype.

## Results

**BMMSCs express Tet proteins**. Since Tet proteins are expressed in various tissues and play an essential biological role in epigenetic regulation, we hypothesized that Tet proteins may affect BMMSC function. We found that both human and mouse BMMSCs express Tet1, Tet2, and Tet3, as assessed by western blotting and real-time polymerase chain reaction (qPCR; Fig. 1a, b). Double immunostaining confirmed that BMMSCs co-express CD146, a mesenchymal stem cell marker, with Tet1, Tet2, and Tet3 (Fig. 1c). It was reported that different Tet proteins may display distinct roles in developmental processes[9]. To explore the possible roles of Tet family members in maintaining BMMSC and bone homeostasis, we used a BMMSC impairment model (ovariectomized (OVX) mice) to assess whether the expression levels of Tet family members were altered in impaired BMMSCs[22]. Micro-computed tomography (micro-CT) and histological analysis confirmed that bone mineral density (BMD), cortical bone area (Ct.Ar), cortical thickness (Ct.Th), and distal femoral trabecular

bone volume of OVX mice were markedly decreased compared with the sham-treated group (Supplementary Fig. 1a–c). The number of colony-forming unit fibroblasts (CFU-F) was significantly elevated in OVX BMMSCs (Supplementary Fig. 1d). Bromodeoxyuridine (BrdU)-labeling assay confirmed that OVX BMMSCs had an increased proliferation rate (Supplementary Fig. 1e). Moreover, OVX BMMSCs showed impaired osteogenic differentiation, as indicated by reduced mineralized nodule formation assessed by alizarin red staining and reduced expression of the osteogenic genes *runt-related transcription factor 2* (*Runx2*), *alkaline phosphatase* (*ALP*), and *osteocalcin* (*OCN*) after 14 days of osteogenic induction, assessed by western blotting (Supplementary Fig. 1f,g). Interestingly, we found that the expression levels of Tet1 and Tet2 were significantly decreased in OVX BMMSCs (Supplementary Fig. 1h). However, the Tet3 expression level remained unchanged. We detected a reduced level of 5-hmC in OVX mice bone marrow stem cells using immunostaining and dot blot assay (Supplementary Fig. 1i-1j), consistent with the fact that Tet proteins convert 5-mC to 5-hmC.

**Tet DKO mice show osteopenia phenotype and BMMSC impairment**. To explore the role of Tet1 and Tet2 in maintaining BMMSC and bone homeostasis, we compared the bone phenotype of $Prx1^{cre}$ (control), $Tet1^{-/-}$, $Prx1^{cre}Tet2^{fl/fl}$, $Tet1^{-/-}$; $Prx1^{cre}Tet2^{fl/fl}$ double knockout (*Tet* DKO) mice at 8–10 weeks of age. Micro-CT and histological analysis showed that *Tet* DKO mice, but not in $Tet1^{-/-}$ mice, had significantly reduced BMD, bone volume/tissue volume (BV/TV), Ct.Ar, Ct.Th, and the distal femoral trabecular bone volume (Fig. 2a–c) compared to littermate controls (Fig. 2a–c). The BMD, BV/TV, Ct.Ar, and the distal femoral trabecular bone volume of *Tet* DKO mice were significantly lower than $Tet1^{-/-}$ and $Prx1^{cre}Tet2^{fl/fl}$ mice, and the bone volume of $Prx1^{cre}Tet2^{fl/fl}$ mice were lower than control group (Fig. 2a–c and Supplementary Fig. 2a). Then, we also performed a calcein-labeling assay to show that the *Tet* DKO mice had a lower bone turn-over rate, which indicated that their bone formation rate was comparatively decreased (Fig. 2d).

To examine whether Tet1 and Tet2 affect BMMSC function, we isolated BMMSCs from 8–10-week-old $Tet1^{-/-}$, $Prx1^{cre}Tet2^{fl/fl}$, *Tet* DKO mice and littermate controls (Supplementary Fig. 2a, b). Flow cytometric analysis showed that BMMSCs from both control and *Tet* DKO mice were positive for stem cell surface markers Sca1, PDGFR, CD105, CD90, and CD73, but were negative for hematopoietic lineage markers CD34 and CD45 (Supplementary Fig. 2c)[23]. The number of CFU-F was significantly elevated in *Tet* DKO but not $Tet1^{-/-}$ and $Prx1^{cre}Tet2^{fl/fl}$ BMMSCs compared to the control group (Fig. 3a). BrdU-labeling assay confirmed that *Tet* DKO but not $Tet1^{-/-}$ and $Prx1^{cre}Tet2^{fl/fl}$ BMMSCs had an increased proliferation rate (Fig. 3b). In addition, we found that *Tet* DKO and $Prx1^{cre}Tet2^{fl/fl}$ BMMSCs but not $Tet1^{-/-}$ BMMSCs showed reduced osteogenic differentiation, as indicated by reduced mineralized nodule formation (Fig. 3c) and decreased expression levels of osteogenic-related genes *Runx2*, *ALP*, and *OCN* (Fig. 3d). In addition, the osteogenic differentiation capacity of $Prx1^{cre}Tet2^{fl/fl}$ BMMSCs was significantly higher than *Tet* DKO BMMSCs (Fig. 3c, d). We further confirmed that BMMSCs derived from *Tet* DKO and $Prx1^{cre}Tet2^{fl/fl}$ mice had significantly reduced capacity for new bone formation when subcutaneously implanted into immunocompromised mice (Fig. 3e). It was reported that Tet1 and Tet2 may compensate for each other[24]. Since the osteopenia phenotype of *Tet* DKO mice and the impairment of *Tet* DKO BMMSCs were significantly severe than that of $Tet1^{-/-}$ and $Prx1^{cre}Tet2^{fl/fl}$ mice, we thus focus on study of *Tet* DKO mice. Control and *Tet* DKO BMMSCs showed similar adipogenic differentiation capacity under adipogenic induction, as

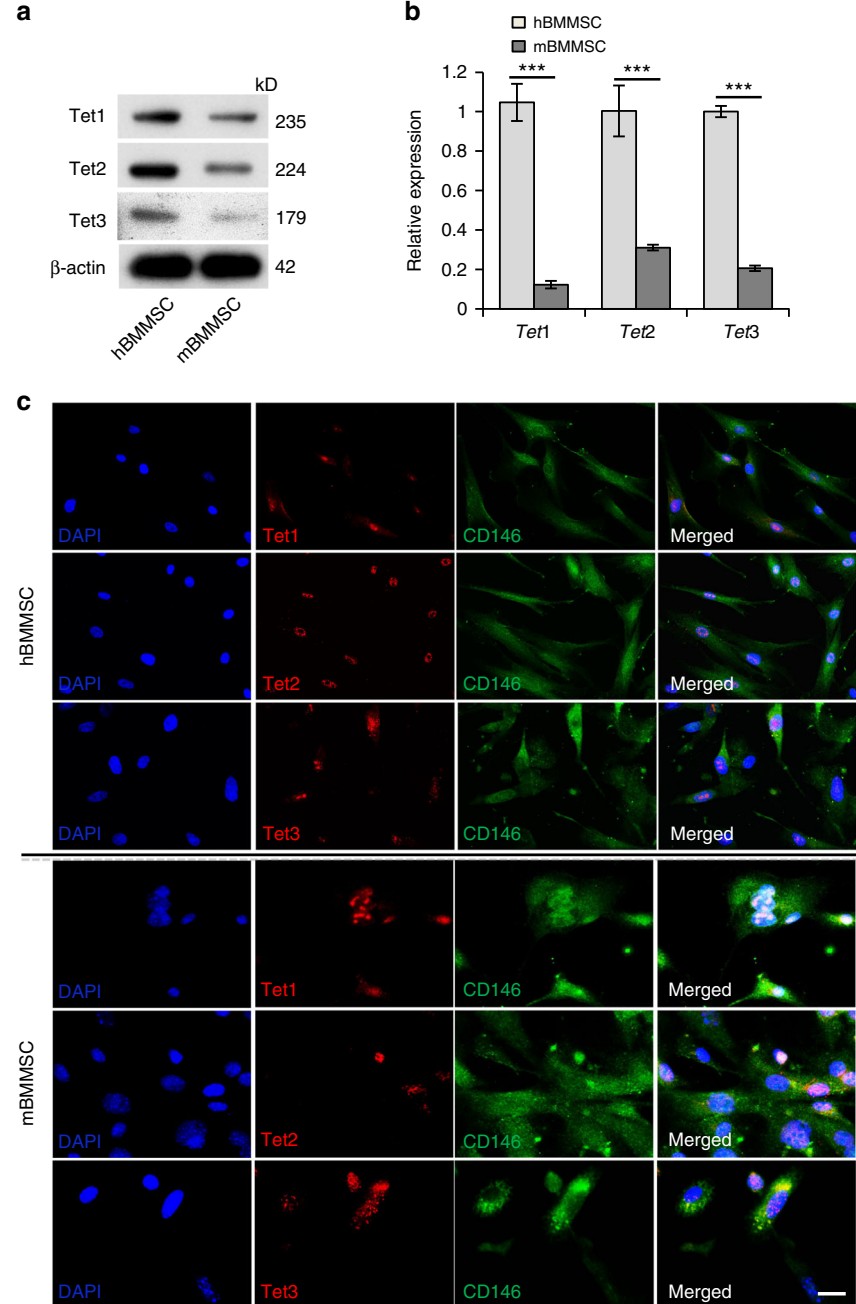

**Fig. 1** BMMSCs express Tet1, Tet2, and Tet3. **a, b** Both human (h) and mouse (m) BMMSCs expressed Tet1, Tet2, and Tet3, as assessed by western blotting (**a**) and qPCR (**b**). **c** Immunocytofluorescent staining showed that CD146-positive BMMSCs expressed Tet1, Tet2, and Tet3. Scale bar, 50 μm. Results are from three independent experiments. ***p <0.001; p values calculated using two-tailed Student's t test (mean ± SD)

indicated by Oil Red O staining and the expression of adipogenic-related genes *lipoprotein* (*LPL*) and *peroxisome proliferator-activated receptor γ2* (*PPARγ2*) (Supplementary Fig. 2d, e). These data indicate that Tet1 and Tet2 depletion impaired BMMSC function.

To further confirm the role of Tet1 and Tet2 in maintaining stem cell properties of BMMSCs in vitro, we used small interfering RNAs (siRNAs) to knockdown Tet1 and Tet2 expression in BMMSCs (Supplementary Fig. 2f). Knockdown of Tet1 and Tet2 in BMMSCs induced a higher proliferation rate (Supplementary Fig. 2g); reduced mineralized nodule formation (Supplementary Fig. 2h); decreased expression levels of osteogenic-related genes *Runx2*, *ALP*, and *OCN* (Supplementary Fig. 2i); and reduced capacity for new bone formation when

subcutaneously implanted into immunocompromised mice (Supplementary Fig. 2j). The elevated proliferation rate and impairment of osteogenic differentiation were consistent with what was observed in Tet1- and Tet2-depleted BMMSCs. These data confirmed that Tet1 and Tet2 are required to maintain BMMSC stem cell properties.

**miRNAs accumulate in *Tet* DKO.** To explore the underlying molecular mechanisms, we performed RNA sequencing (RNA-seq) using RNA from control and Tet1/Tet2 siRNA-treated BMMSCs and found that around 80% of altered genes ($p < 0.05$ and fold change >2) were decreased in Tet1/Tet2 siRNA-treated BMMSCs compared to control group (Supplementary Fig. 3a).

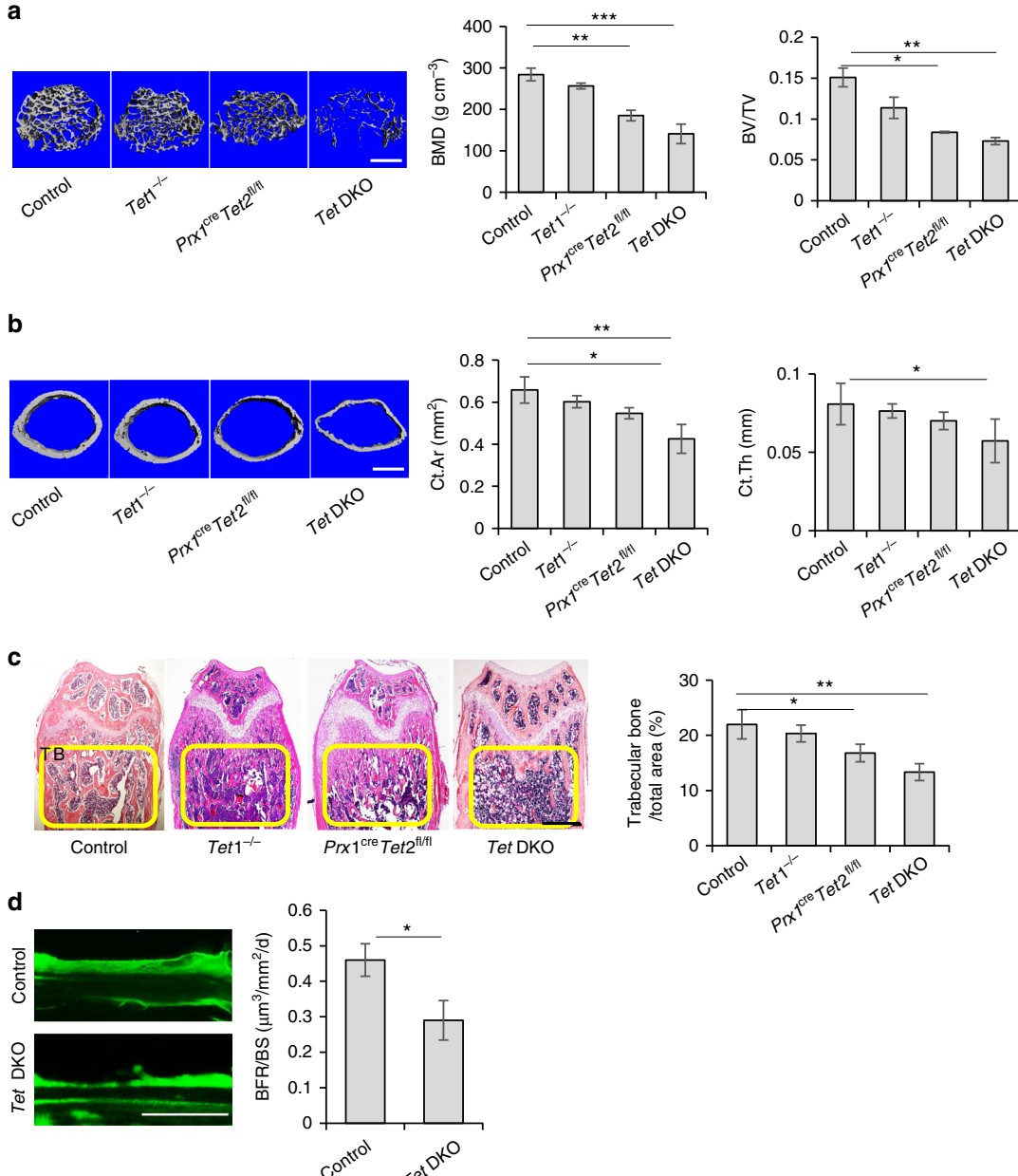

**Fig. 2** *Tet* DKO mice show an osteopenia phenotype. **a** Bone volume/tissue volume (BV/TV) of trabecular bone area in the femurs of control, *Tet1*$^{-/-}$, *Prx1*$^{cre}$*Tet2*$^{fl/fl}$, and *Tet* DKO mice were analyzed by micro-CT. **b** The cortical bone area (Ct.Ar) and cortical thickness (Ct.Th) in the femur of control, *Tet1*$^{-/-}$, *Prx1*$^{cre}$*Tet2*$^{fl/fl}$, and *Tet* DKO mice were assessed by micro-CT. **c** H&E staining showed the trabecular bone volume (yellow-circled area) in the distal femurs of control, *Tet1*$^{-/-}$, *Prx1*$^{cre}$*Tet2*$^{fl/fl}$, and *Tet* DKO mice. **d** Calcein double labeling assay showed the bone formation rate in the metaphyseal trabecular bone of control and *Tet* DKO mice. The 8–10-week-old *Tet1*$^{-/-}$*Prx1*$^{cre}$*Tet2*$^{fl/fl}$ mice were used as *Tet* DKO mice in these experiments, and their littermates whose genetic status was *Prx1*$^{cre}$ were used as controls. *$p < 0.05$, **$p < 0.01$, ***$p < 0.001$ (mean ± SD). Scale bars, 400 μm (**a**, **b**), 1 mm (**c**), and 25 μm (**d**). Results are from three independent experiments. *p* values were calculated using one-way ANOVA (**a-c**) and two-tailed Student's *t* test (**d**)

Functional analysis using WebGestalt showed that 19 of the 40 most significant enriched phenotype categories were related to skeletal bone/cartilage development. These altered genes, including *Runx2*, *ALP*, *Mmp2*, *Msx2*, *Sp7*, and *P2rX7*, were highly related with abnormal skeleton development. *Runx2* is one of the most significant altered genes in all of the 19-phenotype categories (Supplementary Fig. 3b and Supplementary Table 3). Since *Tet* DKO mice displayed a significant osteopenia phenotype and lower bone formation rate in vivo[25, 26], we used western blotting and qPCR analysis to show reduced expression of Runx2 in *Tet* DKO BMMSCs. (Fig. 4a, b). After 14-day osteogenic induction, the expression levels of *Runx2*, *ALP*, and *OCN* were decreased in

*Tet* DKO BMMSCs compared to control group (Fig. 3d). We compared the expression levels of *Runx2*, *ALP*, and *OCN* on different days after osteogenic induction to investigate if *ALP* and *OCN* were potential direct targets or downstream molecules of *Runx2*. The result showed that the expression level of *Runx2*, but not *ALP* and *OCN*, significantly decreased without osteogenic induction and under 3 days of induction (Supplementary Fig. 3c), suggesting that *ALP* and *OCN* may be the downstream target of *Runx2*. There were also less Runx2-positive cells in femur of *Tet* DKO mice compared to control ones (Supplementary Fig. 3d). Next, to evaluate if *Runx2* can be a direct target for Tet1 and Tet2, we analyzed the *Runx2* promoter using Methprimer software and

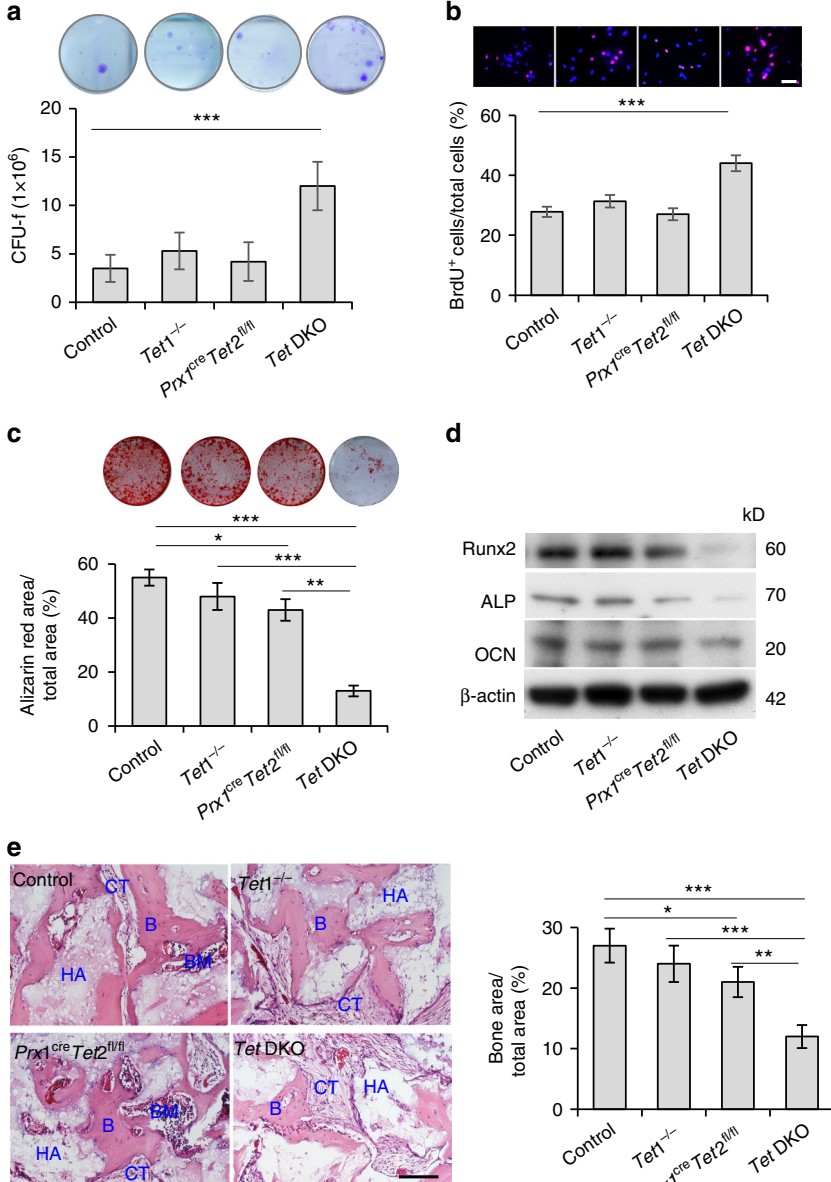

**Fig. 3** *Tet* DKO mice show BMMSC impairment. **a** Toluidine blue staining showed the CFU-F in control, *Tet1*[−/−], *Prx1*[cre]*Tet2*[fl/fl], and *Tet* DKO BMMSCs. **b** BrdU-labeling assay showed the proliferation rate in control, *Tet1*[−/−], *Prx1*[cre]*Tet2*[fl/fl], and *Tet* DKO BMMSCs. **c, d** When cultured under osteogenic inductive conditions, the capacities to form mineralized nodules of control, *Tet1*[−/−], *Prx1*[cre]*Tet2*[fl/fl], and *Tet* DKO BMMSCs were assessed by alizarin red staining (**c**), and the expression of osteogenic markers *Runx2*, *ALP* and *OCN*, as were assessed by western blotting (**d**). **e** New bone (B) formation of control, *Tet1*[−/−], *Prx1*[cre]*Tet2*[fl/fl], and *Tet* DKO BMMSCs when subcutaneously implanted into immunocompromised mice with hydroxyapatite tricalcium phosphate (HA/TCP; HA) as a carrier. *$p < 0.05$, **$p < 0.01$, ***$p < 0.001$ (mean ± SD). Scale bar, 50 μm. Results are from three independent experiments. $p$ values were calculated using one-way ANOVA

found *Runx2* promoter lacks CpG island (Supplementary Fig. 3e). We next used chromatin immunoprecipitation (ChiP)-qPCR to analyze if Tet1/Tet2 could directly binding to the site where CpG was comparably rich in the *Runx2* promoter, and the result showed that no binding site was detected (Supplementary Fig. 3f). Thus, we search potential molecules that may connect Tet with Runx2[27, 28]. Since osteogenic differentiation can be governed by microRNAs (miRNAs) via post-transcriptional regulation[29–31], we investigated whether the variation of miRNAs in *Tet* DKO BMMSCs contributed to the altered *Runx2* expression. We used MicroCosm Targets software to identify 19 miRNAs that may target *Runx2* gene expression (Supplementary Table 1). Interestingly, qPCR analysis showed that the levels of miR-297a-5p,

miR-297b-5p, and miR-297c-5p in *Tet* DKO BMMSCs were significantly higher than in control BMMSCs (Fig. 4c). To further evaluate the functional role of miR-297a-5p, miR-297b-5p, and miR-297c-5p, we used mimics of these miRNAs to treat BMMSCs and inhibitors of these miRNAs to treat *Tet* DKO BMMSCs (Supplementary Fig. 3g, h). *Tet* DKO BMMSCs treated with these miRNA inhibitors showed elevated mineralized nodule formation (Fig. 4d, e) and increased expression levels of *Runx2*, *ALP*, and *OCN* under 14 days of osteogenic induction (Fig. 4f and Supplementary Fig. 3i). Moreover, under osteo-inductive conditions, BMMSCs treated with miR-297a-5p, miR-297b-5p, or miR-297c-5p mimics showed reduced mineralized nodule formation (Fig. 4g) and decreased expression levels of osteogenic-related

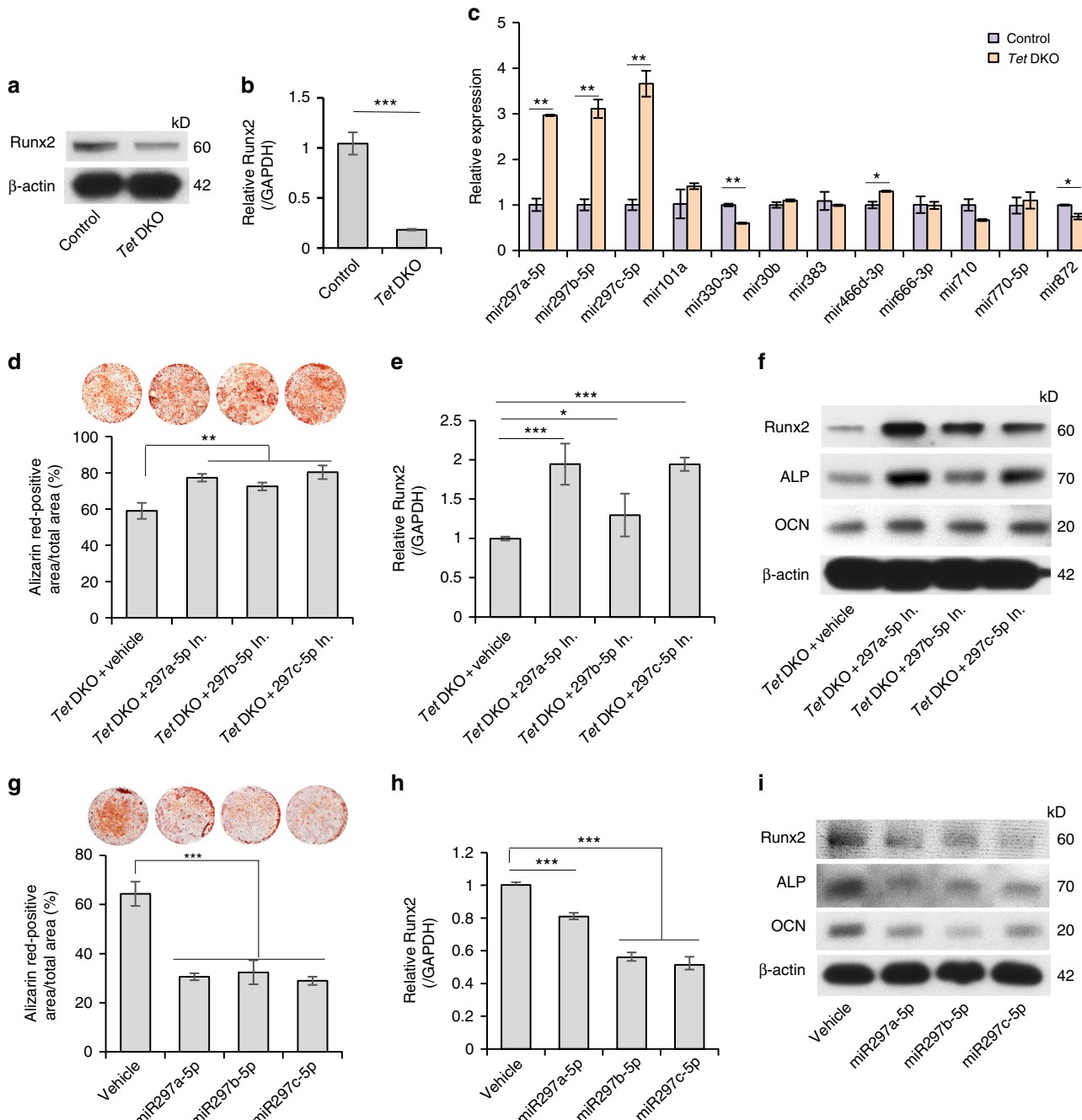

**Fig. 4** miR-297a-5p, miR-297b-5p, and miR-297c-5p accumulate in *Tet* DKO BMMSCs to block *Runx2* signaling. **a**, **b** The expression of *Runx2* in control and *Tet* DKO BMMSCs was analyzed by western blotting and qPCR. **c** The levels of miR-297a-5p, miR-297b-5p, or miR-297c-5p in control and *Tet* DKO BMMSCs assessed by qPCR. **d** Mineralized nodule formation under osteogenic inductive conditions of *Tet* DKO BMMSC after miR-297a-5p, miR-297b-5p, or miR-297c-5p inhibitor treatment. **e**, **f** The expression levels of osteogenic markers *Runx2*, *ALP*, and *OCN* in *Tet* DKO BMMSCs after miR-297a-5p, miR-297b-5p, or miR-297c-5p inhibitor treatment, as assessed by qPCR (**e**) and western blotting (**f**). **g–i** Mineralized nodule formation and the expression of *Runx2*, *ALP*, and *OCN* in BMMSCs after miR-297a-5p, miR-297b-5p, and miR-297c-5p mimic treatment as assessed by alizarin red staining (**g**), qPCR (**h**) and western blotting (**i**). *$p < 0.05$, **$p < 0.01$, ***$p < 0.001$ (mean ± SD). Results are from three independent experiments. $p$ values were calculated using two-tailed Student's $t$ test (**b**, **c**) and one-way ANOVA (**d**, **e**, **g**, **h**)

genes *Runx2*, *ALP*, and *OCN* (Fig. 4h, l and Supplementary Fig. 3j). The proliferation rate of *Tet* DKO BMMSCs was decreased by the miR-297b-5p or miR-297c-5p inhibitor treatment (Supplementary Fig. 3k), while the proliferation rate was significantly upregulated in BMMSCs treated with miR-297b-5p or miR-297c-5p mimics (Supplementary Fig. 3l). Collectively, these data indicate that the accumulation of miR-297a-5p, miR-297b-5p, and miR-297c-5p in *Tet* DKO BMMSCs blocks *Runx2*

signaling and results in impaired osteogenic differentiation and elevated proliferation.

**Tet1 and Tet2 control exosome/miRNA secretion in BMMSCs.** Next, we examined how miR-297a-5p, miR-297b-5p, and miR-297c-5p accumulated in *Tet* DKO BMMSCs. We could not detect CpG island on the promoter of these three miRNAs, indicating

that these may not be directly targeted by Tet1 and Tet2. MSCs can secrete a large number of exosomes[32], which can transfer proteins, mRNAs, and miRNAs to mediate physiological and pathological processes[19, 33, 34]. We therefore sought to test whether exosome secretion is involved in miRNA accumulation in

*Tet* DKO BMMSCs. We found that the amount of exosome-like extracellular vesicles in *Tet* DKO BMMSCs was significantly decreased when compared to the control BMMSCs, as quantified using a Bradford assay (Fig. 5a). We verified that the extracellular vesicles identified in this assay were indeed exosomes, as they

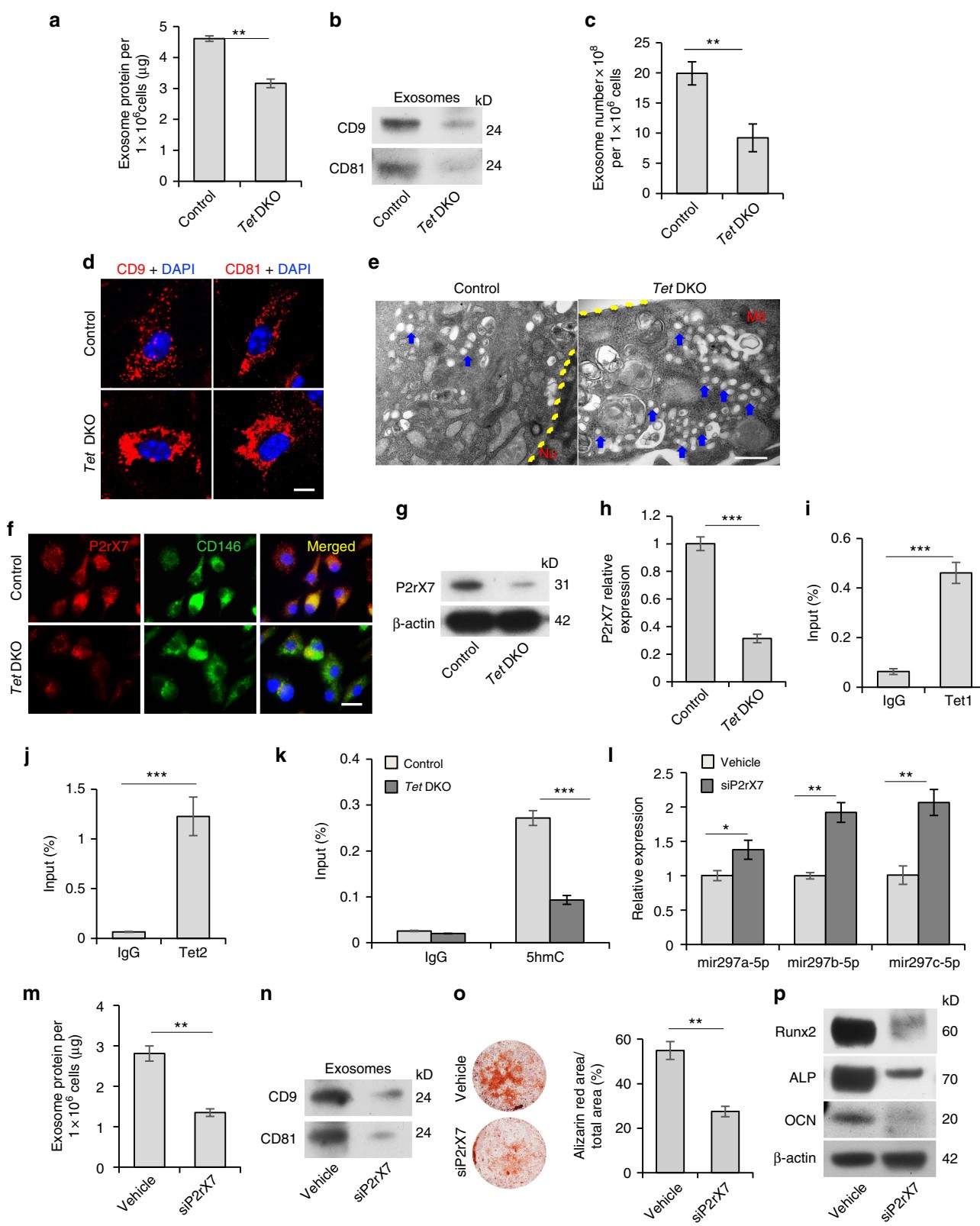

expressed exosome markers CD9 and CD81 (Fig. 5b). Exosomes derived from *Tet* DKO BMMSCs expressed reduced levels of CD9 and CD81 compared to control BMMSCs (Fig. 5b). Furthermore, we used an EXOCEP exosome quantitation kit to show that the exosome secretion was decreased in *Tet* DKO BMMSCs (Fig. 5c)[35]. More intracellular exosomes accumulated in *Tet* DKO BMMSCs compared to control BMMSCs, as indicated by more CD9-positive and CD81-positive intracellular exosomes of *Tet* DKO BMMSCs analyzed by immunofluorescence staining and more vesicle accumulation analyzed by transmission electron microscopy (Fig. 5d, e). Previous studies indicated that *P2rX7* is capable of controlling exosome release[36–38]. Immunocytofluorescence staining showed the co-localization of P2rX7 with MSC surface molecule CD146, along with reduced expression level of P2rX7 in *Tet* DKO BMMSCs (Fig. 5f). We showed that the expression level of P2rX7 was decreased in *Tet* DKO BMMSCs compared to the control group (Fig. 5g, h). In addition, we discovered decreased P2rX7 expression levels in Tet1 and Tet2 siRNA-treated BMMSCs (Supplementary Fig. 4a). Previous studies indicated that Tet1 and 5-hmC prefer to co-localize at transcriptional start sites of CpG-rich promoters[27, 28]. In order to investigate whether Tet1 and Tet2 directly regulate *P2rX7*, we used ChIP-qPCR analysis to show that Tet1 and Tet2 bind to the CpG island of *P2rX7* promoter (Supplementary Fig. 4b, Fig. 5i, j). We next examined whether depletion of Tet1 and Tet2 affected the enrichment of 5-hmC level at the *P2rX7* promoter. Hydroxymethylated DNA immunoprecipitation (hMeDIP)-qPCR analysis revealed that *Tet* DKO BMMSCs showed a remarkably decreased 5-hmC level compared to control BMMSCs (Fig. 5k). Consistently, Tet1 and Tet2 siRNA-treated BMMSCs also displayed reduced 5-hmC enrichment at the *P2rX7* promoter (Supplementary Fig. 4c). MeDIP-qPCR analysis revealed that *Tet* DKO BMMSCs showed increased 5-mC level compared to control BMMSCs (Supplementary Fig. 4d). OxBS sequencing analysis also showed that *Tet* DKO BMMSCs displayed elevated methylation in the promoter of *P2rx7* locus compared to control BMMSCs (Supplementary Fig. 4e). These data suggest that *P2rx7* is a direct target of Tet1 and Tet2 for DNA demethylation. Moreover, we detected the overall level of 5-hmC and 5-mC in BMMSCs by dot blot assay and showed that the level of 5-hmC was lower in *Tet* DKO BMMSCs than the control group, while the level of 5-mC was elevated in *Tet* DKO BMMSCs (Supplementary Fig. 4f, g). Furthermore, we overexpressed wildtype Tet1 and Tet2 plasmid and catalytic domain inactive Tet1 and Tet2 plasmid on *Tet-* DKO BMMSCs to analyze whether overexpression could rescue the decreased expression of P2rX7 and osteogenic differentiation. The results showed that wildtype Tet1 and Tet2 plasmid, but not catalytic domain inactive ones, overexpression rescued decreased expression of P2rX7 in *Tet* DKO

BMMSCs (Supplementary Fig. 4h). Wildtype, but not catalytic domain inactive, Tet1 and Tet2 plasmid overexpression also elevated the mineralized nodule formation and expression levels of *Runx2*, *ALP*, and *OCN* under osteogenic induction (Supplementary Fig. 4h-j).

To further verify the role of *P2rX7* in BMMSC differentiation, we used siRNA to knockdown P2rX7 expression in control BMMSCs (Supplementary Fig. 4k). qPCR analysis showed that siP2rX7-treated BMMSCs had increased levels of intracellular miR-297a-5p, miR-297b-5p, and miR-297c-5p (Fig. 5l). Furthermore, siP2rX7-treated BMMSCs showed a reduced capacity to secrete exosomes into the culture supernatant (Fig. 5m), as well as decreased expression levels of exosome-associated proteins CD9 and CD81 compared to the control BMMSCs (Fig. 5n). Under osteo-inductive conditions, siP2rX7-treated BMMSCs showed reduced mineralized nodule formation (Fig. 5o) and decreased expression levels of osteogenic-related genes *Runx2*, *ALP*, and *OCN* (Fig. 5p). Taken together, these findings suggest that Tet1 and Tet2 are required to maintain BMMSC exosome/miRNA secretion through regulating *P2rX7* demethylation.

### *P2rX7* activation rescues impairment of *Tet* DKO BMMSCs.

Since *P2rX7* deficiency in *Tet* DKO BMMSCs led to the impairment of BMMSC functions, we used a P2rX7 CRISPR activation plasmid to rescue the *P2rX7* level in *Tet* DKO BMMSCs in vitro (Supplementary Fig. 5a). Overexpression of *P2rX7* facilitated the release of accumulated miR-297a-5p, miR-297b-5p, and miR-297c-5p from *Tet* DKO BMMSCs (Fig. 6a) and improved exosome secretion (Fig. 6b). Western blotting showed the expression levels of CD9 and CD81 were also elevated (Fig. 6c). Moreover, osteogenic differentiation of *P2rX7*-overexpressing *Tet* DKO BMMSCs was markedly improved compared to untreated *Tet* DKO BMMSCs, as assessed by alizarin red staining to show elevated mineralized nodule formation (Fig. 6d) and western blotting to show increased expression of the osteogenic genes *Runx2*, *ALP*, and *OCN* (Fig. 6e). We overexpressed P2rX7 in BMMSCs and subsequently implanted into immunocompromised mice subcutaneously with hydroxyapatite/tricalcium phosphate (HA/TCP) as carrier. The result showed that P2rX7 overexpression elevated BMMSC-mediated in vivo new bone formation (Supplementary Fig. 5b).

We next investigated the effect of overexpressing *P2rX7* in 8–10-week-old *Tet* DKO mice by systemic injection of adeno-associated P2rX7 overexpression virus (P2rX7 AAV). P2rX7 AAV or control AAV were injected once a week for up to 4 weeks (Supplementary Fig. 5c). To analyze the efficiency of P2rX7 AAV treatment, we used immunostaining to show the increased *P2rX7* expression in P2rX7 AAV-treated *Tet* DKO mouse bone marrow

**Fig. 5** Tet1 and Tet2 control miRNA secretion in BMMSCs through demethylation of *P2rX7*. **a** Bradford assay showed exosome secretion in control and *Tet* DKO BMMSCs. Exosomes from the culture supernatant of $1 \times 10^6$ BMMSCs containing proteins were assessed. **b** Western blotting showed exosomes from control and *Tet* DKO BMMSCs expressed CD9 and CD81. Exosome proteins from equal volumes of culture supernatant of WT and *Tet* DKO BMMSCs were loaded for western blotting. **c** Exosomes volumes derived from control and *Tet* DKO BMMSCs was detected by EXOCEP exosome quantitation kit. **d** Immunofluorescent staining showed CD9 and CD81-positive red immunofluorescence labeled exosomes localized in control and *Tet* DKO BMMSCs. **e** Transmission electron microscopic (TEM) showed the micro-vesicular in control and *Tet* DKO BMMSCs. **f** *P2rX7* expression (red) was co-localized with CD146 (green) in BMMSCs, as assessed by immunofluorescent double staining. **g**, **h** The expression of *P2rX7* in control and *Tet* DKO BMMSCs, as assessed by western blotting (**g**) and qPCR analysis (**h**). **i**, **j** Tet1 and Tet2 binding on the promoter of *P2rX7* in BMMSCs, as assessed by ChIP-qPCR. IgG was used as a control. **k** Enrichment of 5-hmC in the *P2rX7* promoter in control and *Tet* DKO BMMSCs, as assessed by hMeDIP-qPCR analysis. IgG was used as a control. **l** The levels of miR-297a-5p, miR-297b-5p, and miR-297c-5p in control and *P2rX7* siRNA-treated BMMSCs, as assessed by qPCR. **m** Exosome secretion levels in control and *P2rX7* siRNA-treated BMMSCs, as assessed by Bradford assay. **n** Western blotting showed the expression of CD9 and CD81 in exosomes derived from control and *P2rX7* siRNA-treated BMMSCs. **o**, **p** Mineralized nodule formation and the expression of *Runx2*, *ALP*, and *OCN* in control and *P2rX7* siRNA-treated BMMSCs, as analyzed by alizarin red staining (**o**) and western blot (**p**). *$p < 0.05$, **$p < 0.01$, ***$p < 0.001$ (mean ± SD). Scale bar, 10 μm (**d**), 200 nm (**e**), 50 μm (**f**). Results are from three independent experiments. *p* values were calculated using two-tailed Student's *t* test

cells (Supplementary, Fig. 5d-f). P2rX7 AAV-treated *Tet* DKO BMMSCs were collected and found to have reduced intracellular levels of miR-297a-5p, miR-297b-5p, and miR-297c-5p (Fig. 6f), improved capacity to secrete exosomes into the culture super-natant (Fig. 6g), and elevated expression of CD9 and CD81 (Fig. 6h and Supplementary, Fig. 5g). Micro-CT and histological

analysis showed that P2rX7 AAV-treated *Tet* DKO mice displayed significantly increased BMD, Ct.Ar, Ct.Th, and distal femoral trabecular bone volume compared to control AAV-treated *Tet* DKO mice (Fig. 6i-l). Under osteo-inductive conditions, the osteogenic differentiation of BMMSCs derived from P2rX7 AAV-treated *Tet* DKO mice showed marked

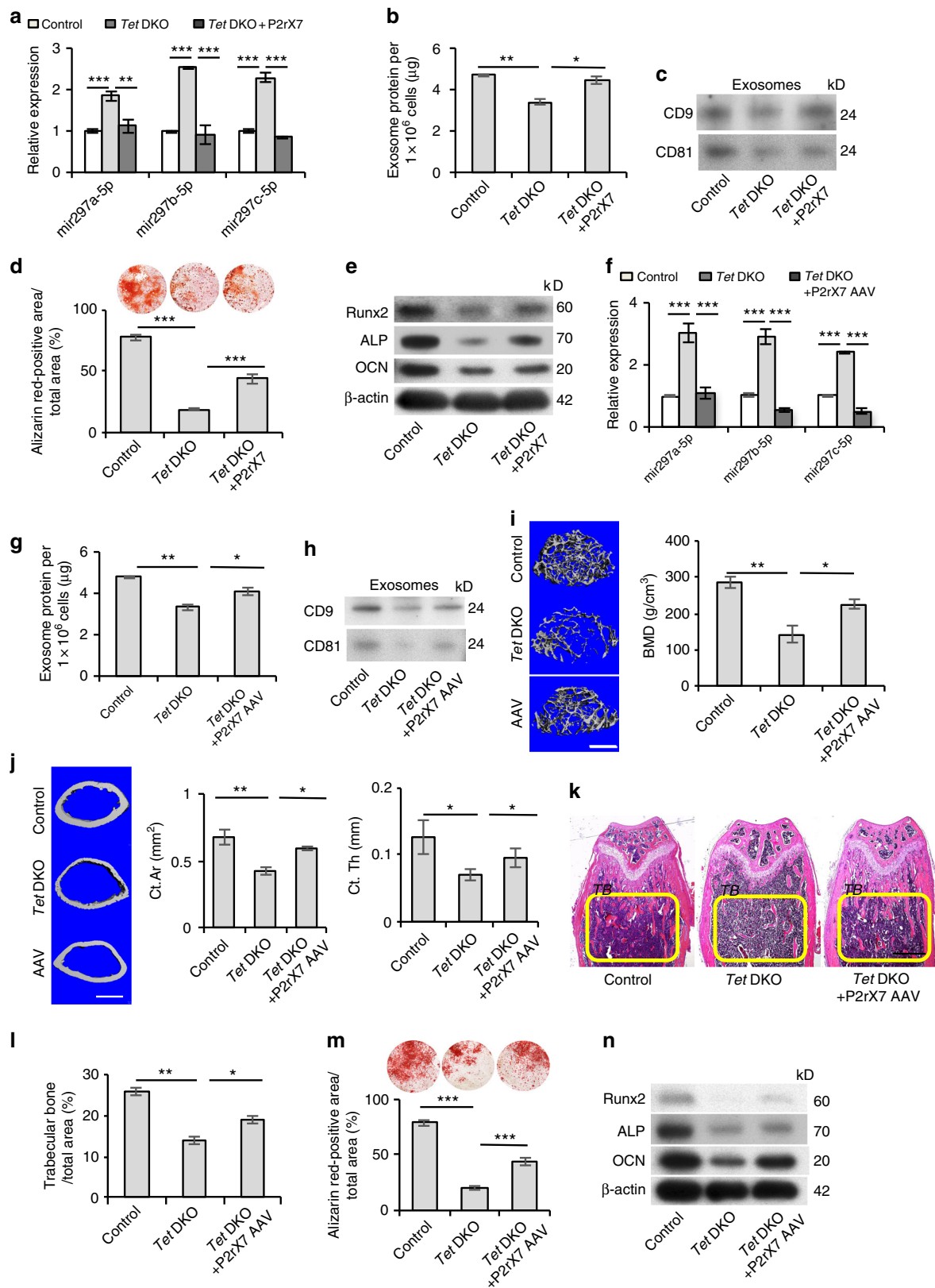

improvement when compared to control AAV-treated *Tet* DKO BMMSCs, as indicated by elevated mineralized nodule formation (Fig. 6m) and increased expression of the osteogenic genes *Runx2*, *ALP*, and *OCN* (Fig. 6n). The cell proliferation rate was decreased after the P2rX7 AAV treatment relative to *Tet* DKO BMMSCs, as assessed by BrdU-labeling assay (Supplementary, Fig. 5h). To verify the role of P2rX7 on bone formation, we used P2rX7 AAV to treat osteopenia (OVX) mice. Micro-CT and histological analysis showed that P2rX7-treated OVX mice had significantly increased BMD, Ct.Ar, Ct.Th, and the distal femoral trabecular bone volume when compared to OVX mice (Supplementary Fig. 6a, c). Moreover, P2rX7 AAV treatment could rescue decreased osteogenic differentiation capacity, but not the elevated proliferation rate in OVX BMMSCs (Supplementary Fig. 6d, f). Collectively, these data indicate that *P2rX7* is capable of regulating intracellular levels of miR-297a-5p, miR-297b-5p, and miR-297c-5p through the control of exosome release. Elevating *P2rX7* levels in *Tet* DKO mice rescued BMMSC function and osteopenia phenotype (Supplementary Fig. 7).

## Discussion

The Tet family contributes to DNA demethylation and acts as a key regulator of epigenetic modulation[1, 2, 5]. In this study, we generated *Tet1*[−/−];*Prx1*[cre]*Tet2*[fl/fl] (*Tet* DKO) conditional knockout mice to reveal the functional role of Tet1 and Tet2 in maintaining BMMSC properties and bone homeostasis. Previous studies showed that loss of Tet2 leads to increased replicating capacity in hematopoietic stem cell and myeloid transformation[39, 40]. Inhibition of Tet1/2 suppresses proliferation of umbilical cord (UC)-derived MSCs[41] and Nanog expression in embryonic stem cells (ES)[1]. Here we showed that Tet1 and Tet2 depletion elevated the proliferation of BMMSCs. Previous study reported that MSCs derived from adipose tissue, BM, and UC origin showed differential pattern of IGF2BP1 expression[41], implying that the different roles of Tet1/2 on stem cell proliferation may be regulated by IGF2BP1 or other molecules. *Tet* DKO BMMSCs exhibited a significant impairment in osteogenic differentiation, which contribute to the osteopenia phenotype observed on *Tet* DKO mice. It was reported that newborns of Tet1 and Tet2 conventional knockout mice did not show developmental delay although a fraction of these double knockout mice were smaller in size[24], suggesting that the osteopenia phenotype observed in *Tet* DKO mice may be attributed to the impairment of BMMSC osteogenic differentiation. Bone homeostasis is a tightly regulated process, balancing new bone formation by BMMSC-derived osteoblasts with bone resorption by osteoclasts. Signals that determine the recruitment, replication, apoptosis, and differentiation of cells of both lineages may increase osteopenia risk[42, 43]. For instance,

alteration of histone H3K9 acetyltransferase PCAF (p300/CBP-associated factor)[44], mammalian target of rapamycin (mTOR) signaling[45] and hydrogen sulfide levels[46] may lead to osteopenia by impairing osteogenic differentiation of BMMSCs. Our previous studies demonstrated that the depletion of Tet1 and Tet2 leads to impairment of regulatory T cell differentiation and immune system disorders[47]. Mutation of Tet2 in HSCs has been reported to impair their function and leads to diverse myeloid malignancies[48]. Except the direct effect of Tet on BMMSCs, whether immune disorder or abnormal of HSCs caused by Tet mutation contributed to bone disorders and BMMSCs impairment need to be further investigated[49].

In this study, we proposed a mechanism by which Tet1 and Tet2 may function as upstream epigenetic regulators of *P2rX7*-controlled exosome release. Exosomes play an important role in intercellular communication because they are able to transfer proteins, mRNAs, and miRNAs[50], contributing to a variety of physiologic processes, including cell homeostasis, differentiation, immune responses, and neuronal signaling[51, 52]. Aberrant exosome secretion may be associated with various human diseases, such as Parkinson's disease, multiple sclerosis, and Alzheimer's disease[52–54]. It is known that multiple microenvironment factors including inflammatory cytokines, hypoxia, and infectious agents are capable of promoting MSC secretion through the activation of an exocytotic pathway[55]. However, whether MSC exosome secretion is modulated by epigenetic modification has remained unclear. In this study, we showed that Tet1 and Tet2 directly bind to the CpG island of *P2rX7* promoter to directly regulate its demethylation activity[36–38]. *P2rX7* is an ionotropic receptor that is controlled by adenosine triphosphate. Activation of *P2rX7* triggers a remarkably diverse range of membrane trafficking responses in leukocytes and epithelial cells[56]. The significance of *P2rX7* in regulating bone development and homeostasis has been demonstrated by previous studies. Bone deficiency was observed in *P2rX7* knockout mice[57, 58]. The combination of this evidence prompted us to further examine the functional role of *P2rX7* in regulating MSC secretion and differentiation. In this study, we demonstrated that the activation of *P2rX7* increased the amount of exosome secretion from *Tet* DKO BMMSCs. *P2rX7* deficiency led to the impairment of osteogenic differentiation of BMMSCs. Collectively, our studies reveal that *P2rX7* is a target of Tet demethylation, and that Tet1 and Tet2 can directly modulate *P2rX7*, thereby controlling BMMSC exosome release to maintain bone and BMMSC homeostasis. Tet1 and Tet2 deficiency also reduced the overall level of 5-hmC in BMMSCs, downregulating expression of other potential molecular targets. Further investigation is required to identify and characterise other downstream target molecules of Tet1 and Tet2 in BMMSCs.

**Fig. 6** *P2rX7* activation rescues impaired *Tet* DKO BMMSCs and osteopenia phenotype in *Tet* DKO mice. **a** The levels of miR-297a-5p, miR-297b-5p, and miR-297c-5p in control, *Tet* DKO BMMSCs, and *P2rX7* CRISPR activation plasmid-treated *Tet* DKO BMMSCs. **b** Bradford assay showed the exosome secretion in control, *Tet* DKO BMMSCs, and *P2rX7* CRISPR activation plasmid-treated *Tet* DKO BMMSCs. **c** Western blotting showed the expression of CD9 and CD81 in control, *Tet* DKO BMMSCs, and *P2rX7* CRISPR activation plasmid-treated *Tet* DKO BMMSCs. **d, e** Mineralized nodule formation and the expression of *Runx2*, *ALP*, and *OCN* in control, *Tet* DKO BMMSCs, and *P2rX7* CRISPR activation plasmid-treated *Tet* DKO BMMSCs, as assessed by alizarin red staining (**d**) and western blotting (**e**). **f** The expression of miR-297a-5p, miR-297b-5p, and miR-297c-5p in control, *Tet* DKO BMMSCs, and adeno-associated *P2rX7* overexpression virus (*P2rX7* AAV)-treated Tet DKO BMMSCs, as assessed by qPCR. **g** Bradford assay showed the exosome secretion level in control, *Tet* DKO BMMSCs, and *P2rX7* AAV-treated BMMSCs. **h** Western blotting showed the expression of CD9 and CD81 in exosomes derived from control, *Tet* DKO BMMSCs, and *P2rX7* AAV-treated BMMSCs. **i** Bone mineral density (BMD) of trabecular bone (TB) area in distal femurs of control, *Tet* DKO, and *P2rX7* AAV-treated-mice, as assessed by micro-CT. **j** The cortical bone area (Ct.Ar) and cortical thickness (Ct.Th) of the femurs of control, *Tet* DKO, and *P2rX7* AAV-treated-mice analyzed by micro-CT. **k, l** H&E staining showed the TB volume (yellow-circled area) of control, *Tet* DKO, and *P2rX7* AAV-treated-mice. **m** The capacity of BMMSCs isolated from control, *Tet* DKO, and *P2rX7* AAV-treated-mice to form mineralized nodules as assessed by alizarin red staining. **n** Western blotting analysis showed the expression levels of the osteogenic markers *Runx2*, *ALP*, and *OCN* in control, *Tet* DKO, and *P2rX7* AAV-treated *Tet* DKO BMMSCs. The 8–10-week-old *Tet1*[−/−]*Prx1*[cre]*Tet2*[fl/fl] mice were used as *Tet* DKO mice in our experiments, and their littermates whose genetic status were *Prx1*[cre] were used as controls. *$p < 0.05$, **$p < 0.01$, ***$p < 0.001$ (mean ± SD). Scale bars, 400 μm (**i, j**), 1 mm (**k**). Results are from three independent experiments. *p* values were calculated using one-way ANOVA

The miRNAs are small non-coding RNAs with a length of 18–23 nucleotides that function as post-transcriptional regulators of gene expression[59]. Herein, we computationally identified 19 miRNAs that may target *Runx2*. *Tet* DKO BMMSCs showed a different miRNA expression profile than that of control BMMSCs. Previous studies showed that miR-29b, miR-26a, and miR-767 can target Tet1 to inhibit its expression, leading to a change in 5-hmC levels[60–62]. Conversely, the Tet family can also directly modulate miR-200 by demethylating the CpG islands of the miR-200 promoter[63]. In this study, as there is no CpG island on the promoter of miR-297a-5p, miR-297b-5p, and miR-297c-5p, we reveal a mechanism of interaction between the Tet family and miRNAs and show that Tet1 and Tet2 depletion leads to *P2rX7* promoter hypermethylation and reduced exosome secretion. This results in intracellular accumulation of miR-297a-5p, miR-297b-5p, and miR-297c-5p in *Tet* DKO BMMSCs. In turn, the intracellular accumulation of miR-297a-5p, miR-297b-5p, and miR-297c-5p decreases the osteogenic differentiation of *Tet* DKO BMMSCs by inhibiting *Runx2* signaling. Interestingly, we also found that the proliferation rate of BMMSCs is elevated after miR-297b-5p and miR-297c-5p mimic treatment, which is consistent with our findings on *Tet* DKO BMMSCs. Previous studies indicated that the biological role of *Runx2* is not restricted to activating osteoblast lineage-specific genes; it also contributes to negative control of cell proliferation[64]. Our finding may help to explain why the *Runx2*-deficient in *Tet* DKO BMMSCs and the intracellular accumulation of *Runx2*-targeting miRNAs results in the impairment of osteogenic differentiation and elevated proliferation rate. Overexpression of *P2rX7* rescues the exosome release and accumulation status of miR-297a-5p, miR-297b-5p, and miR-297c-5p in *Tet* DKO BMMSCs. In vivo overexpression of *P2rX7* rescues the osteopenia phenotype and BMMSC function in *Tet* DKO mice, suggesting that *P2rX7* may be a potential therapeutic target for treating Tet-associated skeletal disorders. Based on the previous studies, deletion of Tet proteins might lead to other pathogenic process other than osteopenia[65]. Thus, the effect of the exosome secretion impairment caused by Tet depletion might have a broad effect on miRNA release, which may not only specify to the miRNAs targeting Runx2. Global gene analysis showed *Runx2* cluster genes were altered after Tet1 and Tet2 knockdown in BMMSCs, implying that the Tet/P2rX7/ Runx2 cascade may be one of critical regulating mechanism to maintain bone homeostasis and BMMSC function. As 5-hmC may be involved in establishing and maintaining chromatin structure for both actively transcribed genes and PcG-repressed regulators, it also affects both transcriptional activation and repression roles in a context-dependent manner. The detailed mechanisms of how 5-hmC regulates other genes altered by Tet in BMMSCs may need further investigation[66, 67].

In conclusion, depletion of Tet1 and Tet2 results in BMMSC impairment and an osteopenia phenotype. An increase in exosome secretion and reduction of intracellular accumulation of miR-297a-5p, miR-297b-5p, and miR-297c-5p, achieved by restoring the level of *P2rX7* in *Tet* DKO BMMSCs, can rescue BMMSC function and bone homeostasis.

## Methods

**Animals**. Female C57BL/6J (JAX #000664), B6.129S4-Tet1$^{tm1.1Jae}$/J (JAX #017358, *Tet1*$^{+/−}$), B6.129S-Tet2$^{tm1.1Iaai}$/J (JAX #017573, *Tet2*$^{fl/fl}$), and B6.Cg-Tg (Prrx1-cre) 1Cjt/J (JAX #005584, *Prx1*$^{cre}$) mice were purchased from the Jackson Laboratory (Bar Harbor, ME, USA). Age-matched littermates were used in all experiments. The 8–10-week-old C57Bl/6J mice were ovariectomized (OVX)[22], and sham-operated age-matched female mice served as controls. We purchased immuno-compromised nude mice (Beige *nu.nu* XIDIII) from Harlan (Indianapolis, IN, USA). To generate *Tet1*$^{−/−}$*Prx1*$^{cre}$*Tet2*$^{fl/fl}$ mice, we mated *Tet1*$^{+/−}$*Prx1*$^{cre}$*Tet2*$^{fl/+}$ mice with *Tet1*$^{+/−}$*Tet2*$^{fl/+}$ mice, and littermates whose genetic status was *Prx1*$^{cre}$ were used as controls. All animal experiments were performed under institutionally

approved protocols for the use of animal research (University of Pennsylvania Institutional Animal Care and Use Committee #805478).

**Antibodies**. The antibodies to Tet1 (ab191698), Tet2 (ab94580), P2rX7 (ab48871), CD146 (ab24577), PDGFRα (ab65258), and OCN (ab10911) were purchased from Abcam (Cambridge, MA, USA). Antibodies to ALP (sc-28904), CD9 (sc-9148), and CD81 (sc-9158) were purchased from Santa Cruz Biotechnology Inc. (Santa Cruz, CA, USA). Antibody to Runx2 (8486) was obtained from Cell Signaling Technology (Danvers, MA, USA). Antibody to Tet3 (20602) was purchased from Novus Biologicals (Littleton, CO, USA). Anti-CD34-PE (551387) and SCA1-PE (553108) antibodies were purchased from BD Biosciences (San Jose, CA, USA). Anti-CD105-PE (12-1051-82), CD45-PE (25-0454-82), CD73-PE (12-0739-42), and CD90-PE (15-0902-82) antibodies were purchased from eBioscience (San Diego, CA, USA). Anti-β-Actin antibody was purchased from Sigma-Aldrich (St. Louis, MO, USA).

**Isolation of mouse BMMSCs**. We collected femurs and tibias from mice and flushed out the bone marrow cells with 2% fetal bovine serum (FBS; Equitech-Bio, Kerrville, NY, USA) in phosphate-buffered saline (PBS). All nuclear cells (ANCs) were seeded ($15 \times 10^6$ cells per dish) in 100 mm culture dishes (Corning, Tewsburg, MA, USA) and incubated at 37 °C under 5% $CO_2$ conditions. After 48 h, non-adherent cells were washed by PBS and adherent cells were cultured in alpha minimum essential medium (α-MEM, Invitrogen, Grand Island, NY, USA) supplemented with 20% FBS, 2 mM L-glutamine (Invitrogen), 55 μM 2-mercaptoethanol (Invitrogen), 100 μg ml$^{−1}$ streptomycin, and 100 U ml$^{−1}$ penicillin (Invitrogen) for an additional 14 days. Mouse BMMSCs in passage one were used in this study. For CFU-F assay, we seeded $1 \times 10^6$ ANCs in 60 mm cell culture dishes (Corning). The cells were washed with PBS and stained with 1% toluidine blue solution with 2% paraformaldehyde (PFA, Sigma-Aldrich) after 16 days of culturing. Clusters with more than 50 cells were counted as colonies under microscopy.

**Isolation and culture of human BMMSCs**. We purchased human bone marrow from AllCells LLC (Alameda, CA, USA), which were aspirated from healthy human adult volunteers (20–35 years of age). The dissociated cell suspension was filtered through a 70 μm cell strainer, then cultured with α-MEM containing 10% FBS, 10 mM L-ascorbic acid phosphate, 100 U ml$^{−1}$ penicillin, 100 μg ml$^{−1}$ streptomycin, and 2 mM L-glutamine, incubated at 37 °C under 5% $CO_2$ condition. After 48 h, the non-adherent cells were removed. The adherent cells were passaged with 0.05% trypsin containing 1 mM EDTA after being cultured for 14 days.

**Immunofluorescent staining**. We seeded BMMSCs on chamber slides (Nunc, Rochester, NY, USA) ($2 \times 10^3$/well) and fixed the cells with 4% PFA. The chamber slides or histological sections were incubated with primary antibodies (1:100) at 4 °C overnight, then treated with Alexafluoro 488 or Alexafluoro 568 conjugated secondary antibody (1:200, Invitrogen) for 1 h at room temperature. Finally, we mounted the slides with Vectashield mounting medium containing 4′,6-diamidino-2-phenylindole (DAPI; Vector Laboratories, Burlingame, CA, USA).

**Western blotting**. Total protein was lysed in M-PER mammalian protein extraction reagent (Thermo, Rockford, IL, USA). After the proteins (20 μg) were loaded and separated to 4–12% NuPAGE gel (Invitrogen), they were transferred to 0.2 μm nitrocellulose membranes (Millipore, Billerica, MA, USA). Then, 0.1% Tween-20 and 5% non-fat dry milk were provided to block the membranes for 1 h, followed by overnight incubation with primary antibodies (1:1000) diluted in blocking solution. The membranes were washed and incubated for 1 h in horse-radish peroxidase (HRP)-conjugated secondary antibody (Santa Cruz Biotechnology; 1:10,000). SuperSignal West Pico Chemiluminescent Substrate (Thermo) and BioMax film (Kodak) were used to detect the immunoreactive proteins. All full blots presented in the main figures are displayed in the Supplementary Information.

**Real-time PCR**. We used miRNeasy Mini Kit (Qiagen, Valencia, CA, USA) to isolate total RNA from the cultured cells according to the manufacturer's instructions. For qPCR of mRNA, we used SuperScript III Reverse Transcriptase (RT) kit (Invitrogen) to prepare the complementary DNA (cDNA). qPCR was performed using SYBR Green Supermix (Bio-Rad, Hercules, CA, USA) and gene-specific primers. The level of mRNA expression for each gene was normalized to glyceraldehyde 3-phosphate dehydrogenase (GAPDH). For qPCR of miRNA, miScript II RT Kit (Qiagen) was used to synthesize the cDNA. miScript SYBR Green PCR Kit (Qiagen) was used to perform real-time PCR. We used RUN6 as an endogenous control for BMMSCs. A CFX96 Real-Time PCR System (Bio-Rad) was used for qPCR analysis.

**RNA-sequencing**. RNA-seq analyses of total RNA from vehicle and Tet1 and Tet2 siRNA-treated BMMSCs were performed at the Department of Biotechnology, Beijing Institute of Radiation Medicine. Three RNA samples from each group were used for RNA-seq analysis. For each sample, we used NEBNext Ultra RNA library

Pre Kit to prepare a sequencing library from 1 μg of total RNA, and 2 × 100 paired-end sequencing in fat run mode was performed using the HiSeq 2500 and Illumina TruSeq SBS-Kit v2 (200 cycles).

**Bioinformatics analysis.** Illunina's bcl bcl2fastq v1.8.4 software (Illumina,San Diego, CA) was used to convert the resulting base calling (.bcl) to FASTQ files. Mapping RNA-seq reads on the mouse genome (*Mus musculus* GRCm38) after trimming the adaptors, transcript assembly, and abundance estimation were performed using DNASTAR Lasergene v15.0 (DNASTAR, Madison, WI) and reported using FPKM (fragments per kilobase of exon per pillion fragments mapped). The Gene Pattern and WebGestalt were used for functional analysis.

**In vitro osteogenic differentiation.** BMMSCs were cultured in osteogenic medium containing 100 μM L-ascorbic acid 2-phosphate (Wako), 2 mM β-glycerophosphate (Sigma-Aldrich), and 10 nM dexamethasone (Sigma-Aldrich) in the growth medium. After 14 days of osteogenic induction, western blotting was performed to detect the expression levels of Runx2, ALP, and OCN. After 4 weeks of induction, mineralized nodule formation was detected by staining with 1% Alizarin Red S (Sigma-Aldrich) for the cultures. The stained positive areas were quantified using the National Institutes of Health (NIH) ImageJ software and shown as a percentage of the total area.

**Cell proliferation assay.** Mouse BMMSCs ($10 \times 10^3$/well) were cultured on 2-well chamber slides (Nunc, Rochester, NY, USA) for 2–3 days. After being incubated with BrdU solution (1:100) (Invitrogen) for 20 h, the cultures were stained with BrdU antibody (1:200, Invitrogen) at 4 °C overnight, then treated with Alexafluoro 568 conjugated secondary antibody for 1 h at room temperature. Finally, Vecta-shield mounting medium containing DAPI (Vector Laboratories, Burlingame, CA, USA) was used to mounted slides. BrdU-positive cells were counted and indicated as a percentage of the total cell number, and 10 images were analyzed per subject. Three independent samples were repeated for each experimental group.

**Isolation and characterization of exosomes.** Exosomes were isolated from BMMSCs by ultracentrifuge. Exosome-depleted medium (complete medium depleted of FBS-derived exosomes by overnight centrifugation at $100,000 \times g$) was used to culture cells for 48 h. We used differential centrifugation to extract exosomes from culture supernatants of $12 \times 10^6$ BMMSCs[68], at $300 \times g$ for 10 min, $3000 \times g$ for 10 min, $20,000 \times g$ for 30 min, and $120,000 \times g$ for 70 min. For analysis of exosome protein secretion, Bradford protein assay (Bio-Rad Laboratories) was used to measure the amount of total protein in purified exosomes from $12 \times 10^6$ BMMSCs. Total exosome protein was normalized to $1 \times 10^6$ cells to show the amount as micrograms per $1 \times 10^6$ cells. EXOCEP exosome quantitation kit (System Biosciences Inc.) was used to quantitate the number of vesicles, following the manufacturer's instruction[35].

**siRNA knockdown, CRISPR activation plasmid, Tet plasmid microRNA mimic, and inhibitor transfection.** For siRNA knockdown, we seeded $0.2 \times 10^6$ BMMSCs on a 6-well culture plate. Tet1, Tet2, and P2rX7 siRNAs (Santa Cruz Biotechnology) were used to treat the BMMSCs according to the manufacturer's instructions. Non-targeting control siRNAs (Santa Cruz Biotechnology) were used as negative controls. pCDF-mTET1 (Addgene #81052), pCDF-His-mTET1CDΔcat (Addgene #81054), pcDNA3-Tet2 (Addgene #60939), and pScalps_Puro_mTet2 catalytic domain HxD (Addgene #79611) were purchased from Addgene (Cambridge, MA). For P2rX7 CRISPR activation and Tet plasmid (Santa Cruz Biotechnology) transfection, we seeded $0.2 \times 10^6$ BMMSCs on a 6-well culture plate, and transfected with P2rX7 CRISPR activation plasmid (Santa Cruz Biotechnology) using Lipofectamine LTX with Plus reagent (Life Technologies) according to the manufacturer's instructions. We used control CRISPR activation plasmids as a negative control. The miR-297a-5p, miR-297b-5p, and miR-297c-5p mimics, inhibitors and negative controls (Qiagen) were transfected into cells according to the manufacturer's instructions. Briefly, BMMSCs ($0.2 \times 10^6$) were seeded on a 6-well culture plate and transfected with miR-297a-5p, miR-297b-5p, miR-297c-5p mimics, or inhibitors using Lipofectamine LTX with Plus reagent (Life Technologies) according to the manufacturer's instructions.

**Methylated and hydroxymethylated DNA immunoprecipitation (hMeDIP).** Immunoprecipitation of 5-mC and 5-hmC was performed using Active motif MeDIP kit and hMeDIP Kit with minor modifications. We used Branson sonicator to sonicate DNA into short fragments (100 to 1000 base pairs (bp)) for 20 min with 15 s on, 15 s off cycles at low power. We heat-denatured sonicated DNA at 95 °C for 10 min. Sonicated DNA (1 μg) was immunoprecipitated with 4 μl of mouse anti-5-methylcytosine or anti-5-hydroxymethylcytosine monoclonal antibody (Active Motif, $1 \mu g \mu l^{-1}$). After overnight incubation at 4 °C, we added magnetic beads to the DNA-antibody mixture and samples were incubated for 2 h at 4 °C. Isolation of immunoprecipitated DNA was performed according to the kit instructions. SYBR® Green Supermix (Bio-Rad) was used to perform qPCR on a Bio-Rad CFX96 Real Time system, as indicated by the manufacturer's protocol.

The percentage of enrichment was calculated relative to the amount of DNA used in the IP reaction.

**Chromatin immunoprecipitation-quantitative PCR (ChIP-qPCR).** $5 \times 10^6$ cells were cross-linked and used for each immunoprecipitation, and chromatin was sheared with a Branson sonicator. Millipore ChIP kit was used to perform immunoprecipitation according to the manufacturer's protocol. To precipitate DNA-protein complexes, Tet1 and Tet2 antibodies were used and IgG was used as an isotype control. Percentage input was determined by removing an aliquot of sheared chromatin prior to immunoprecipitation and comparing amplification of this DNA to amplification of the precipitated chromatin. Putative Tet1 and Tet2 binding sites in the P2rX7 promoter were investigated using MethPrimer software to predict the CpG island[69].

**Transmission electron microscopy.** For transmission electron microscopy (TEM), control and *Tet* DKO BMMSCs were collected by centrifugation and fixed with 2.5% glutaraldehyde overnight and then fixed with 1% osmium tetraoxide for 2 h, dehydrated in a graded series of ethanol concentrations, and embedded in SPIPON812 resin and polymerized. The block was sectioned by microtome (Leica EM UC6). The ultrathin sections approximately 70 nm, mounted on copper grids, uranyl acetate, and lead citrate were used to stain, and examined, and photographed with a FEI Tecnai spirit TEM (FEI Tecnai Spirit 120 kv).

**Histology.** To assess trabecular bone and bone marrow areas, femurs were fixed in 4% PFA, decalcified with 5% EDTA (pH 7.4), and embedded in paraffin. For histological analysis, sections were deparaffinized and stained with hematoxylin and eosin (H&E). The trabecular bone area percentage was then calculated using NIH ImageJ software.

**Micro-CT analysis.** Femurs were fixed in 4% PFA, and analyzed using a μCT scanner (micro-CT 35, Scanco Medical AG, Brüttisellen, Switzerland). Briefly, the distal end of the femur corresponding to a 0 to 4.1 mm region above the growth plate was scanned at 6 μm isotropic voxel size. For the trabecular bone analysis, the images of the secondary spongiosa regions 0.6 to 1.8 mm above the highest point of the growth plate were contoured. BMD and BV/TV were calculated using standard 3D microstructural analysis. After smoothing the image with a 3D gaussian low-pass filter (Sigma 0.7 voxels), the CT images were segmented into bone and marrow regions by applying a visually chosen for the cortical bone assessment. The outer and inner perimeter of the cortical midshaft was determined by a 3D tri-angulation of the bone surface of the 20 slices, and the formula Ct.Th = 1/2 × BS/BV was used to calculate cortical thickness. Ct.Ar was calculated by the formula Ct.Ar = thickness of ring × length of middle line = thickness × (outer circumference + inner circumference)/2.

**P2rX7 AAV treatment in *Tet* DKO mice.** P2rX7 overexpression adeno-associated virus (P2rX7 AAV, Applied Biological Materials Inc., Richmond, BC, Canada) was used to overexpress P2rX7 in vivo. P2rX7 AAV ($1 \times 10^9$ GC) or AAV control virus was injected into the *Tet* DKO mice once a week for up to 4 weeks. After the therapy, samples were harvested immediately for histological assessments and BMMSC isolation.

**In vivo BMMSC-mediated bone formation.** We subsequently implanted $4.0 \times 10^6$ BMMSCs mixed with 40 mg HA/TCP ceramic powder (Zimmer Inc., Warsaw, IN, USA) under the dorsal surface of 8-week-old immunocompromised mice. The transplants were harvested at 8 weeks post implantation, fixed in 4% PFA, and decalcified with 5% EDTA, then embedded in paraffin. The sections were stained with H&E. For quantification of in vivo new bone regeneration, we calculated the area of bone formation with 10 representative images from different regions of the BMMSC implants using ImageJ software (National Institutes of Health, Bethesda, MD). Each experimental group was repeated with three independent implants.

**Dot blot assay.** DNA was diluted to 100 ng ml$^{-1}$ and a dilution series was performed; 1 μl was spotted on a 0.45 μm pore size positively charged nylon membrane and wait for drying in the air, then the membrane was baked in 80 °C vacuum oven for 2 h. The membrane was blocked using 5% milk/0.1%Tween-20/Tris-buffered saline for 1 h, and being incubated with 5-hmC (1:1000) or 5-mC (1:1000) antibodies overnight at 4 °C. Then, the membrane were washed and incubated for 1 h in HRP-conjugated secondary antibody (Santa Cruz Biotechnology; 1:10,000). Finally, SuperSignal West Pico Chemiluminescent Substrate (Thermo) and BioMax film (Kodak) were used to do the detection.

**Methylation-specific PCR and OxBS sequencing.** To assess P2rX7 methylation status, optimized oxidative bisulfite sequencing (OxBS sequencing) was used. A highly selective chemical oxidizes 5-hmC to 5-formylcytosine (5fC) and then bisulfite treatment deformylated and deaminated 5fC to uracil, and appears as thymine (T) in sequencing analysis. The 5-mC is not deaminated and appears as a cytosine (C) in sequencing analysis. Oxidative bisulfite conversion, which is able to

produce an accurate readout distinguishing 5-mC from 5-hmC, was performed using a TrueMethyl™ kit (Cambridge Epigenetix, UK), according to the manufacturer's instructions[70]. For methylation-specific PCR, pretreated DNA was amplified with methylation-specific primers for P2rX7 as listed in the supplementary table 2 and were sequencing as we previously reported[47].

**Statistics and reproducibility.** For animal study, we used at least five mice and experiments were performed three times, unless otherwise stated in the figure legends. No specific statistical test was used to predetermine the sample size. Comparisons between two groups were analyzed using independent unpaired two-tailed Student's *t*-tests, and comparisons between more than two groups were analyzed using one-way analysis of variance with the Bonferroni correction if the data did not met the normality distribution assumption. The *p* values of less than 0.05 were considered statistically significant. In vitro and in vivo experiments were studied in a non-blinded fashion, no method of randomization was used, and typically no sample was excluded for the analysis.

**Data availability.** The data that support the findings in this study are available from the corresponding author upon reasonable request. RNA-sequencing data have been deposited in the Gene Expression Omnibus (GEO) database by GSE108872.

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

## Acknowledgements

This work was supported by grants from the National Institute of Dental and Craniofacial Research, National Institutes of Health, Department of Health and Human Services (R01DE017449 to S.S. and K99DE025915 to C.C.), International Science & Technology Cooperation Program of China (2015DFB30040 to S.S. and Y.Z.), an NIH NIAMS fellowship (T32AR007442), the Young Elite Scientist Sponsorship Program by CAST (2017QNRC001), Beijing Natural Science Foundation (7182182), and an Schoenleber Pilot Research Grant from University of Pennsylvania School of Dental Medicine. We would like to thank the Center for Biological Imaging, the Institute of Biophysics, for our electron microscopy work and we would be grateful to Lei Sun and Can Peng for their help of making EM samples and taking EM images. We would like to thank the Department of Biotechnology, Beijing Institute of Radiation Medicine, for our RNA-sequencing work and we would be grateful to Zhe Zhou for his help for doing RNA-sequencing and bioinformatics analysis.

## Author contributions

S.S. and Y.Z. conceived the project. S.S., R.Y., T.Y., and Y.Z. designed the experiments. R. Y., T.Y., C.C., X.G., X.K., and D.L. performed the experiments and analyzed the data. S.S. and T.Y. wrote the manuscript.

## Additional information

**Competing interests:** The authors declare no competing interests.

