## [Peer Review File · Nature Communications]

Reviewers' comments:

Reviewer #1 (Remarks to the Author):

In this study, Yang and coworkers showed the effect of Tet family on maintaining BMMSC and bone homeostasis. The study provides thorough evidence that Tet 1 and Tet 2 epigenetically regulate BMMSC through P2rx7 to control exosome and the release of miR-297a-5p, miR-297b-5p and miR-297c-5p. They also investigated the role of Tet/P2rx7/Runx2 in treating osteopenia disorders. This study would benefit by clarification of the following points:

Major issues:

1. Tet family is critical for self-renewal and differentiation of hematopoietic stem cells (HSCs). Clinically, somatic loss-of-function mutations of TET2 are frequently observed in patients with diverse myeloid malignancies, including myelodysplastic syndromes, myeloproliferative neoplasms, and chronic myelomonocytic leukemia, not bone disorders. Therefore, it is interesting to know whether the osteopenia phenotype of Tet DKO is caused by MSCs instead of HSCs.
2. In result 1 (Line 18), please justify ovariectomized (OVX) mice using as a BMMSC impairment model.
3. The study showed that 75% of the newborn Tet1-depletion mice exhibit a smaller body size. Do Tet DKO mice have any phenotype on cortical bone?
3. In Figure 6, systemic overexpression P2rx7 as a treatment of BMMSCs impairment and bone defect may not be an ideal therapy as P2rx7R is broadly expressed in immune cells of the hematopoietic lineage including monocytes, lymphocytes, macrophages, and dendritic cells.

Minor issues:

1. The kDa of all protein in WB analysis should be provided.
2. In Figure S1g, the quantification of protein level should be provided.
3. In Figure S1h, it's better to co-stain 5hmC with stem cell markers because the study wanted to confirm that Tet proteins convert 5-mC to 5-hmC in BMMSCs.
4. In Figure 5B, K, and M, the quality of Western-blot should improve.
5. The age of mice should be mentioned in every part of results.

Reviewer #2 (Remarks to the Author):

The manuscript by Yang et al. is an interesting study that reports that TET1 and 2 maintain adult MSC homeostasis and hence bone regeneration in postnatal mice by regulating P2rx7, a regulator of exosome function.

While the study tackles a significant issue of the role of the TET enzymes in skeletal regeneration, the central weakness is that the underlying mechanism for the TET function is underdeveloped and not well supported by the experiments. For the conclusions to be solid, there are multiple major issues that need to be clarified by further experimentation.

Major concerns-

1. Although the study focuses on P2rx7 as the major TET1/2 target , which is also utilized for the rescue of the osteopenia phenotype in the TET1/2 DKO mouse-its relationship with TET1/2 is unclear. If a broad cellular function such as exosome secretion is inhibited upon P2rx7 downregulation, there can be secreted factors that affect cell-cell communication during bone formation and an overexpression of P2rx7 could be beneficial irrespective of the TET1/2 defect. (a) It would therefore be important to check if the AAV mediated P2rx7 would be beneficial even in a TET1/2 wild-type mouse upon ovariectomy, to establish its link with TET1/2 as their bonafide target. (b) Is the effect of the exosome inhibition on the miRNA retention expected...this would be a very broad effect though and not just restricted to the MiRNAs targeting Runx?

2. Although the study focuses on P2rx7 as the major TET1/2 target...both the mechanism for P2rx7 regulation by TETs is weak and the expts do not rule out Runx as a direct target for the TETs. The epigenetic analyses of the P2rx7 as well as Runx are needed along with the exact transcription (there is no mRNA data for Runx2 at all). A bisulfite and TAB-sequencing based base resolution analysis of the promoter and enhancer elements of the 2 putative targets is needed to understand the precise effects of Tet1 and 2 loss.

It needs to be tested if TET1/2 bind to Runx2 promoter.

It is also not explained if the effect on ALP and OCN is direct or downstream of Runx2-all the experimentation appears to be at late stages of osteogenic differentiation of BMMSC (3 weeks or so) but testing early stages will help discern the direct effect on Runx2 expression from its downstream targets.

Also, the precise effects of TET1 or 2 can be discerned through a knockdown approach (si or shRNA) that will complement the genetic mice experiments. For example, it is possible that the loss of TET2 alone in the mesenchyme can recapitulate all of the effects shown in the study-these points need to be tested and clarified.

3. Although the mouse model is referred to as TET1/2 Double knockout i.e. TET DKO throughout, the mouse is actually a total KO for TET1 with the TET2 being knocked out in the mesenchyme through PRX-cre. It is not clear what 'control' is utilized throughout. It would be helpful to label the precise genetic status rather than just 'control'. This calls for multiple controls to correctly assess the precise effects of TET1 and 2 that are largely missing in all experiments- the TET1 KO, TET2 KO through Prx alone. This is especially pertinent since the TET1 global KO has been shown to have some skeletal defects (Dawlaty et al 2011, Dev Cell).

4. Another major concern is that the isolation of BM-MSc by just concentrating for the cells that stick to plastic would lead to a very heterogenous population-there is no FACS sorting or even just a single cell FACS based analyses to assess both the stemness and the heterogeneity of the populations. This is a major issue with all the invitro experiments with the bone marrow derived MSCs. Sca1 and PDGFR have been utilized previously to isolate mouse MSC. See below- <http://www.nature.com/nprot/journal/v7/n12/full/nprot.2012.125.html?elq=b1a4061545a347c88a84111633b44f05>

5. The osteopenia phenotype after OVX in wild-type mice is very prominent compared to the TETDKO mouse-the authors should characterize OVX in the mutant mouse to differentiate potential developmental defects that the KO mice harbor from strictly postnatal MSC based bone regeneration.

6. Exosome characterization is very weak-FACS based analysis using CD9 and CD81 is required. The exosome isolation method also needs further details and some quality control to assess the enrichment of exosomes and the variation in preparation across samples. Present data with the western blot is of poor quality and needs quantitation and an independent technique to verify.

Minor concerns-

1. How are Runx2/Alp/Ocn regulated in the wildtype OVX model upon osteopenia induction-what are the 5mC/5hmC/C dynamics at their regulatory regions.
2. Loss of 5hmC in WT ovx (sup Fig 2) not convincing-a quantitative ELISA assay is needed.
3. How was the BrdU labeling done and how are the BMMSC populations isolated and analyzed for these assays? It is important to analyze whether the proliferation change is limited to/due to a subpopulation.
4. Fig 4d- quantitation needed since the data does not convincingly show an effect of the miRNA inhibitors on Runx expression-why is the transcription data not collected/analyzed?
5. Is the effect of miRNAs restricted to Runx since a change in Alp/Ocn always accompanies Runx downregulation. Inhibitor treatment data on Day 0 and early time points of osteogenic differentiation is needed to assess a direct and specific effect on Runx 2.

Reviewer #3 (Remarks to the Author):

Yang et al report that inhibition of Tet1 and Tet2 results in impaired bone differentiation of bone marrow MSCs (BMMSCs) and a significant osteopenia phenotype. The authors further claim that Tet1 and Tet2 deficiency reduces demethylation of the P2rx7 promoter and thus downregulates exosome release, leading to intracellular accumulation of miR-297a-5p, miR-297b-5p, and miR-297c-5p and these miRNAs inhibit Runx2 signaling to impair BMMSC function. The manuscript seems to provide a novel understanding of how BMMSCs regulate normal skeletal homeostasis and should be of great interest to the readers; however, the authors need to provide further evidence to support their conclusion.

Major points:

1. The authors claim that Tet1 and Tet2 inhibition reduces demethylation of the promoter of P2rx7 that regulates exosome release, leading to intracellular accumulation of miR-297a-5p, miR-297b-5p, and miR-297c-5p. To support their conclusion, they need to demonstrate that (a) exosomes released from control MSCs contain miR-297a-5p, miR-297b-5p, and miR-297c-5p and (b) treatment of these exosomes derived from control MSCs, but not the Tet DKO MSC-derived exosomes, suppresses osteogenic differentiation in control MSCs. (c) In addition, it would be important to show that Tet knockdown does not affect adipogenic differentiation potential in BMMSCs. Therefore, the authors need to show the result of adipogenic differentiation of either Tet siRNA transfected BMMSCs or Tet DKO mouse derived BMMSCs compared to their control BMMSCs. (d) Furthermore, I think it would be important to confirm these key observations with human BMMSCs as the authors mention that the Tet/P2rx7/Runx2 cascade may serve as a target for the development of novel therapies for osteopenia disorders.
2. The authors claim that depletion of Tet1 and Tet2 impairs self-renewal and differentiation of BMMSCs as they observed that Tet DKO mouse derived BMMSCs show an increase in CFU-f and BrdU positive cells in Figures 3a and 3b while showing the reduced bone differentiation. However, since Mahaira et al (PMID: 24915579) previously reported the opposite result that inhibition of TET1/2 suppresses proliferation and Nanog levels in umbilical cord (UC)-derived MSCs, it would be

important to provide strong evidence to support their conclusion. Previously, Moran-Crusio et al showed that Tet2 loss leads to increased hematopoietic stem cell self-renewal and myeloid transformation (PMID: 21723200). Hence, if the early passage BMMSC cultures contain the transformed hematopoietic cells, it could affect the results of CFU-f, BrdU and differentiation assays. Indeed, mouse MSCs isolated by the classical method of plastic adherence are frequently contaminated by overgrown hematopoietic cells [PMID: 10022616]. Since the authors used passage one mouse MSCs for the current study, it would be important to demonstrate that the increased CFU-f and BrdU positive cells and decreased osteogenic differentiation in Tet DKO BMMSC culture are not due to contamination of the transformed hematopoietic cells. Therefore, MSC characterization data including cell morphology, adipogenic differentiation and flow cytometry analysis for surface epitopes of MSC markers (CD90, CD105 and CD73) and hematopoietic lineage markers (CD34 and CD45) need to be provided.

3. They further claim that Tet DKO mice show a significant osteopenia phenotype. However, Tet DKO seems to have a marginal effect on bone density in mice based on the results in Figure 2. To support their conclusion, the authors need to provide more convincing in vivo data. In addition, they need to show a proof of knockout of the target genes in transgenic mice.

4. The authors claim that P2rx7 AAV increases secretion of exosomes in Tet DKO BMMSCs based on their western blot assay result in Figure 6h. However, the authors need to provide quantitative data of the western blotting results with statistical analysis since I don't see much difference in CD9 levels in Figure 6h.

Minor points:

The authors need to provide the following information;

- Sequence of primers for real-time PCR
- A cell source of human BMMSCs
- A method of BrdU labeling assay
- An explanation of how new bone formation was quantitated in Figure 3e and supplementary Figure 2e.
- Information of primary antibodies for immunofluorescence staining
- What does BFR / BS stand for in Figure 2c

Reviewer #4 (Remarks to the Author):

The study by Shi and colleagues focuses on the role of Tet1 and Tet2 in BMMSC biology and its implications in osteopenia. They find that Tet1 and Tet2 are expressed in BMMSC and contribute to 5hmC levels. They show that loss of Tet1 and Tet2 in BMMSCs in mice leads to increased osteopenia. Mechanistically they show that Runx2 levels are down regulated in BMMSC due to increased levels of inhibitory microRNAs that accumulate in the cell because of impaired exosome release. They find silencing of P2rx7 gene, which is a mediator of exosome release, and establish that Tet1/2 is responsible for proper hydroxylation of P2rx7 promoter and its expression.

While the study is carefully carried out, it only focuses on one selected pathway. If the goal of the study is to define the role of Tet enzymes in BMMSC biology then more in depth and broader approaches should be considered. As it is, it does not comprehensively address the role of Tet enzymes in BMMSC biology as the authors propose to achieve in this study.

Main points.

1. What is the level of expression of Tet enzymes in BMMSC relative to other tissues or stem cell

types such as embryonic or neural stem cells?

2. What are the overall levels of 5hmC in BMMSCs and how much of it is contributed by Tet1 and Tet2? Immunofluorescence images are not very clear and do not provide quantitative data. Moreover, how does loss of Tet-mediated hydroxylation influence 5mC levels?

3. How does loss of Tet1 and Tet2 influence the global gene expression programs in BMMSCs? Are there other mechanisms other P2rx7 and Runx2 involved? Some form of global gene expression analyses would provide better insight.

4. How does changes in hydroxylation affect methylation in BMMSCs and how does that influence gene expression? I understand that the authors focus only on hydroxylation of P2rx7, but loss of these enzymes can influence many genes and loci globally with possible direct or indirect implications in BMMSC properties.

5. The in vivo phenotype, and also the rescue results, though significant, are very marginal. The authors should comment on this as to how such small changes influence BMMSC and contribute to increased osteopenia.

6. If the mechanism is through hydroxylation of P2rx7 promoter the authors should show if it is the hydroxylation or the subsequent demethylation that allows for proper expression of P2rx7. I think 5mC levels at promoter should be quantified too. Also locus specific sequencing for 5hmC and 5mC should be done instead of DIP-PCR.

7. I do not see a rescue of phenotypes by Tet1 or Tet2 re-expression. This would be useful, especially if catalytic mutant versions of Tet enzymes are tested too.

8. Are the levels of miRNAs increased because of exosome release issue only? Or that Tet enzymes can regulate their levels directly as the authors mention in the discussion.

Other points:

1. I do not find clear images showing that the impaired release of exosome leads to increased number of exosomes inside the cells. They only show less exosomes released.

2. The deletion efficiency of Tet2 floxed allele is not shown. Also the ages at which the analyses are done are not clearly documented.

3. The cartoon depicting the model is very misleading. There should be side-by-side models for presence and absence of Tet1/2 to clearly illustrate the story.

Reviewers' comments:

Reviewer #1 (Remarks to the Author):

In this study, Yang and coworkers showed the effect of Tet family on maintaining BMMSC and bone homeostasis. The study provides thorough evidence that Tet 1 and Tet 2 epigenetically regulate BMMSC through P2rx7 to control exosome and the release of miR-297a-5p, miR-297b-5p and miR-297c-5p. They also investigated the role of Tet/P2rx7/Runx2 in treating osteopenia disorders. This study would benefit by clarification of the following points:

Major issues:

1. Tet family is critical for self-renewal and differentiation of hematopoietic stem cells (HSCs). Clinically, somatic loss-of-function mutations of TET2 are frequently observed in patients with diverse myeloid malignancies, including myelodysplastic syndromes, myeloproliferative neoplasms, and chronic myelomonocytic leukemia, not bone disorders. Therefore, it is interesting to know whether the osteopenia phenotype of Tet DKO is caused by MSCs instead of HSCs.

Response: We appreciate the reviewer's concern. In this study, we used Tet1 knockout and Tet2 mesenchymal tissue conditional knockout (*Tet1^{-/-}; Prx1^{cre} Tet2^{fl/fl}*) mice. Thus, the osteoporosis phenotype observed in these Tet DKO mice is caused by BMMSC impairment instead of HSC alteration. We have showed the mice background information in the Method part and added the following sentences in the Discussion section on page 15: "Tet2 mutation in hematopoietic stem cells (HSCs) may impair their function and lead to diverse myeloid malignancies (Tefferi and others, 2009) (Delhommeau and others, 2009). It is unknown whether Tet2 deficiency-caused HSC impairment contributes to bone disorders and BMMSCs impairment, it needs to be further investigated (Wang and others, 2013)."

2. In result 1 (Line 18), please justify ovariectomized (OVX) mice using as a BMMSC impairment model.

Response: We thank the reviewer for the suggestion. We have made a justification accordingly in the Result section on page 4.

3. The study showed that 75% of the newborn Tet1-depletion mice exhibit a smaller body size. Do Tet DKO mice have any phenotype on cortical bone?

Response: We thank the reviewer for this valuable comment. In order to examine the cortical bone phenotype, we harvested the femurs of *Tet* DKO mice for micro-CT analysis. We found that Tet1 and Tet2 depletion resulted in reduced cortical bone area (Ct.Ar) and cortical thickness (Ct.Th). We have added these data in the Results section on page 5: “we compared the bone phenotype of *Prx1^{cre}* (control), *Tet1^{-/-}*, *Prx1^{cre}Tet2^{fl/fl}*, and *Tet1^{-/-};Prx1^{cre}Tet2^{fl/fl}* double knockout (*Tet* DKO) mice at 8-10 weeks of age. Micro-CT and histological analysis showed that *Tet* DKO mice, but not *Tet1^{-/-}* mice, had significantly reduced bone mineral density (BMD), bone volume/tissue volume (BV/TV), cortical bone area (Ct.Ar), cortical thickness (Ct.Th) and distal femoral trabecular bone volume (Fig. 2a-2c) compared to littermate controls (Fig. 2a and 2b). The BMD, BV/TV, Ct.Ar and the distal femoral trabecular bone volume of *Tet* DKO mice were significantly lower than those of *Tet1^{-/-}* and *Prx1^{cre}Tet2^{fl/fl}* mice, and the bone volume of *Prx1^{cre}Tet2^{fl/fl}* mice was lower than that of control group (Fig. 2a-2c and Supplementary Fig.2a).”

3. In Figure 6, systemic overexpression P2rX7 as a treatment of BMMSCs impairment and bone defect may not be an ideal therapy as P2rX7R is broadly expressed in immune cells of the hematopoietic lineage including monocytes, lymphocytes, macrophages, and dendritic cells.

Response: We appreciate the reviewer for this thoughtful suggestion. P2rX7 has been reported to be expressed in several cell types including immune, neuronal and bone cells (Lenertz and others, 2011). The functional role of P2rX7 in bone formation and osteogenic stem cells has also been reported (Agrawal and others, 2017; Li and others, 2015). Here, to avoid potential off-target effect of systemic overexpression of P2rx7, we overexpressed P2rX7 in BMMSCs and subsequently implanted them into immunocompromised mice subcutaneously. We have added these data in the Results section on page 12: “We overexpressed P2rX7 in BMMSCs and subsequently implanted them into immunocompromised mice subcutaneously with HA/TCP as a

carrier. The results showed that P2rX7 overexpression elevated BMMSC-mediated *in vivo* new bone formation (Supplementary Fig. 5b).”

Minor issues:

1. The kDa of all protein in WB analysis should be provided.

Response: We have labeled the kDa for all the proteins in the Western blot analysis.

2. In Figure S1g, the quantification of protein level should be provided.

Response: We have repeated this Western blot analysis and quantified the protein level.

3. In Figure S1h, it's better to co-stain 5hmc with stem cell markers because the study wanted to confirm that Tet proteins convert 5-mC to 5-hmC in BMMSCs.

Response: We agree with this excellent suggestion, so we co-stained the 5hmC with mesenchymal stem cell surface marker CD146 and replaced image Fig S1h.

4. In Figure 5B, K, and M, the quality of Western-bolt should improve.

Response: We have replaced the Western blot data in Fig. 5B, 5K, and 5M with newly generated data.

5. The age of mice should be mentioned in every part of results.

Response: The age of mice was added to every experiment described in the Results.

Reviewer #2 (Remarks to the Author):

The manuscript by Yang et al. is an interesting study that reports that TET1 and 2 maintain adult MSC homeostasis and hence bone regeneration in postnatal mice by regulating P2rX7, a regulator of exosome function.

While the study tackles a significant issue of the role of the TET enzymes in skeletal regeneration, the central weakness is that the underlying mechanism for the TET function is underdeveloped and not well supported by the experiments. For the conclusions to be solid, there are multiple major issues that need to be clarified by further experimentation.

Response: We appreciated the reviewer's positive comments.

Major concerns-

1. Although the study focuses on P2rX7 as the major TET1/2 target, which is also utilized for the rescue of the osteopenia phenotype in the TET1/2 DKO mouse-its relationship with TET1/2 is unclear. If a broad cellular function such as exosome secretion is inhibited upon P2rX7 downregulation, there can be secreted factors that affect cell-cell communication during bone formation and an overexpression of P2rX7 could be beneficial irrespective of the TET1/2 defect.

(a) It would therefore be important to check if the AAV mediated P2rX7 would be beneficial even in a TET1/2 wild-type mouse upon ovariectomy, to establish its link with TET1/2 as their bonafide target.

Response: We thank the reviewer for the thoughtful suggestion. We overexpressed P2rX7 in wild-type ovariectomized (OVX) mice. Micro-CT and histological analysis showed that P2rX7-treated OVX mice had significantly increased bone mineral density (BMD), cortical bone area (Ct.Ar), cortical thickness (Ct.Th) and distal femoral trabecular bone volume compared to OVX mice. P2rX7 overexpression could rescue the decreased osteogenic differentiation capacity of BMSCs in OVX mice (Supplementary Fig. 6). We have added these newly generated data in the Results section on page 13: “To verify the role of P2rX7 in bone formation, we used P2rX7

AAV to treat osteopenia (OVX) mice. Micro-CT and histological analysis showed that P2rX7-treated OVX mice had significantly increased bone mineral density (BMD), cortical bone area (Ct.Ar), cortical thickness (Ct.Th), and distal femoral trabecular bone volume when compared to OVX mice (Supplementary Fig. 6a-6c). Moreover, P2rX7 AAV treatment could rescue the decreased osteogenic differentiation capacity, but not the elevated proliferation rate in OVX BMSCs (Supplementary Fig. 6d-6f).”

(b) Is the effect of the exosome inhibition on the miRNA retention expected...this would be a very broad effect though and not just restricted to the MiRNAs targeting Runx?

Response: We appreciate the reviewer for this thoughtful suggestion. Impaired exosome release in BMSCs might affect the accumulation of various functional miRNAs, as our previous study showed non-Runx2-related miRNAs were regulated by exosomes released by MSCs (Liu and others, 2015). We compared the global gene expression of control and Tet1/Tet2 siRNA-treated BMSCs using RNA-sequencing. The newly generated data were added to the Results section on pages 7-8: “To explore the underlying molecular mechanism, we performed RNA-sequencing analysis using RNA from control and Tet1/Tet2 siRNA-treated BMSCs and found that around 80% of altered genes ($p < 0.05$ and fold change > 2) were decreased in Tet1/Tet2 siRNA-treated BMSCs compared to the control group (Supplementary Fig. 3a). Functional analysis using WebGestalt showed that 19 of the 40 most significant enriched phenotype categories were related to skeletal bone/cartilage development. These altered genes, including *Runx2*, *ALP*, *Mmp2*, *Msx2*, *Sp7*, and *P2rX7*, are strongly associated with abnormal skeleton development. *Runx2* is one of the most significantly altered genes in all of the 19 phenotype categories (Supplementary Fig. 3b and Supplementary Table 3).”

In this study, we focus on miRNAs that target on Runx2. There may be miRNAs that target other genes in different contexts. We have therefore added the following sentences in the Discussion on page 17: “Based on the previous studies, deletion of Tet proteins might lead to other pathogenic processes other than osteopenia (An and others, 2017). Thus, the exosome secretion impairment caused by Tet depletion might have a broad effect on miRNA release, which may not be specific to the miRNAs targeting Runx2. Global gene analysis showed *Runx2* cluster genes were altered after Tet1 and Tet2 knockdown in BMSCs, implying that the Tet/P2rX7/Runx2 cascade may be one of the critical regulating mechanisms for maintaining bone homeostasis and

BMMSC function.”

2. Although the study focuses on P2rX7 as the major TET1/2 target...both the mechanism for P2rX7 regulation by TETs is weak and the exps do not rule out Runx as a direct target for the TETs. The epigenetic analyses of the P2rX7 as well as Runx are needed along with the exact transcription (there is no mRNA data for Runx2 at all). A bisulfite and TAB-sequencing based base resolution analysis of the promoter and enhancer elements of the 2 putative targets is needed to understand the precise effects of Tet1 and 2 loss. It needs to be tested if TET1/2 bind to Runx2 promoter.

Response: We thank the reviewer for the valuable suggestions. In order to define if Runx2 is a direct target for Tet1 and Tet2, we analyzed the promoter of *Runx2* using Methprimer software and Chip-qPCR. We added the data in the Result section on Page 8: “Next, to evaluate whether Runx2 may be a direct target for Tet1 and Tet2, we analyzed the *Runx2* promoter using Methprimer software and found that it lacks a CpG island (Supplementary Fig. 3e). We next used ChiP-qPCR to analyze whether Tet1/Tet2 could directly binding to the site where CpG is comparatively rich in the *Runx2* promoter. The result showed that no binding site was detected (Supplementary Fig. 3f). Thus, we searched for potential molecules that may connect Tet with Runx2 (Kohli and Zhang, 2013; Xu and others, 2011).”

Moreover, we have added the *Runx2* mRNA data in Fig. 4 accordingly.

To demonstrate the effect of Tet1 and Tet2 on P2rX7, we added the ChiP-qPCR analysis for 5mC and Ox-BS sequencing analysis for the methylation status of *P2rX7* promoter to the Results section on page 11: “MeDIP-qPCR analysis revealed that *Tet* DKO BMMSCs showed increased 5mC levels compared to control BMMSCs (Supplementary Fig. 4d). Ox-BS sequencing analysis also showed that *Tet* DKO BMMSCs displayed elevated methylation in the promoter of *P2rx7* locus compared to control BMMSCs (Supplementary Fig. 4e).”

It is also not explained if the effect on ALP and OCN is direct or downstream of Runx2-all the experimentation appears to be at late stages of osteogenic differentiation of BMMSC (3 weeks or so) but testing early stages will help discern the direct effect on Runx2 expression from its downstream targets.

Response: We thank the reviewer for this valuable suggestion. To investigate whether ALP and OCN are direct targets or otherwise downstream molecule of *Runx2*, we measured the *Runx2*, *ALP*, and *OCN* expression levels on the early and later osteogenic differentiation stages. We have added the newly generated data in the Result section on page 8: “After 14 days of osteogenic induction, the expression levels of *Runx2*, *ALP* and *OCN* were decreased in *Tet* DKO BMMSCs compared to the control group (Fig 3d). We compared the expression levels of *Runx2*, *ALP*, and *OCN* on different days after osteogenic induction to investigate whether *ALP* and *OCN* are potential direct targets or otherwise downstream molecules of *Runx2*. The result showed that the expression level of *Runx2*, but not *ALP* and *OCN*, significantly decreased in non-osteogenic induction or after 3 days of induction conditions (Supplementary Fig. 3c), suggesting that *ALP* and *OCN* maybe the downstream targets of *Runx2*.”

Also, the precise effects of TET1 or 2 can be discerned through a knockdown approach (si or shRNA) that will complement the genetic mice experiments. For example, it is possible that the loss of TET2 alone in the mesenchyme can recapitulate all of the effects shown in the study-these points need to be tested and clarified.

Response: We appreciate the reviewer’s suggestion. To dissociate the precise effects of Tet1 and Tet2, we analyzed the phenotypes of *Tet1*^{-/-} and *Prx1*^{Cre}*Tet2*^{fl/fl} mice. Newly generated data were added to the Results section on page 5: “..., we compared the bone phenotype of *Prx1*^{cre} (control), *Tet1*^{-/-}, *Prx1*^{cre}*Tet2*^{fl/fl}, and *Tet1*^{-/-};*Prx1*^{cre}*Tet2*^{fl/fl} double knockout (*Tet* DKO) mice at 8-10 weeks of age. Micro-CT and histological analysis showed that *Tet* DKO mice, but not *Tet1*^{-/-} mice, had significantly reduced bone mineral density (BMD), bone volume/tissue volume (BV/TV), cortical bone area (Ct.Ar), cortical thickness (Ct.Th) and distal femoral trabecular bone volume (Fig. 2a-2c) compared to littermate controls (Fig. 2a and 2b). The BMD, BV/TV, Ct.Ar and the distal femoral trabecular bone volume of *Tet* DKO mice were significantly lower than those of *Tet1*^{-/-} and *Prx1*^{cre}*Tet2*^{fl/fl} mice, and the bone volume of *Prx1*^{cre}*Tet2*^{fl/fl} mice was lower than that of control group (Fig. 2a-2c and Supplementary Fig.2a).”

The following sentences were also added to the Results section on page 6: “It has been suggested that Tet1 and Tet2 may compensate for each other (Dawlaty and others, 2013). Since the osteopenia phenotype of *Tet* DKO mice and the impairment of *Tet* DKO BMMSCs were

significantly more severe than those of *Tet1*^{-/-} and *Prx1*^{cre}*Tet2*^{fl/fl} mice, thus we focus primarily on *Tet* DKO mice.”

[Redacted]

Parts of this peer review file have been redacted as indicated.

[Redacted]

3. Although the mouse model is referred to as TET1/2 Double knockout i.e. TET DKO throughout, the mouse is actually a total KO for TET1 with the TET2 being knocked out in the mesenchyme through PRX-cre. It is not clear what ‘control’ is utilized throughout. It would be helpful to label the precise genetic status rather than just ‘control’. This calls for multiple controls to correctly assess the precise effects of TET1 and 2 that are largely missing in all experiments- the TET1 KO, TET2 KO through Prx alone. This is especially pertinent since the TET1 global KO has been shown to have some skeletal defects (Dawlaty et al 2011, Dev Cell).

Response: We thank the reviewer for the valuable comments. We generated $Tet1^{-/-}Prx1^{cre}Tet2^{fl/fl}$ mice to serve as *Tet* DKO mice. Their littermates with phenotype $Prx1^{cre}$ were used as controls. We compared the phenotypes of $Prx1^{cre}$, $Tet1^{-/-}$, $Prx1^{cre}Tet2^{fl/fl}$ and $Tet1^{-/-}Prx1^{cre}Tet2^{fl/fl}$ (*Tet* DKO) mice and the data were added to the Results section on page 5: "..., we compared the bone phenotype of $Prx1^{cre}$ (control), $Tet1^{-/-}$, $Prx1^{cre}Tet2^{fl/fl}$, and $Tet1^{-/-};Prx1^{cre}Tet2^{fl/fl}$ double knockout (*Tet* DKO) mice at 8-10 weeks of age. Micro-CT and histological analysis showed that *Tet* DKO mice, but not $Tet1^{-/-}$ mice, had significantly reduced bone mineral density (BMD), bone volume/tissue volume (BV/TV), cortical bone area (Ct.Ar), cortical thickness (Ct.Th) and distal femoral trabecular bone volume (Fig. 2a-2c) compared to littermate controls (Fig. 2a and 2b). The BMD, BV/TV, Ct.Ar and the distal femoral trabecular bone volume of *Tet* DKO mice were significantly lower than those of $Tet1^{-/-}$ and $Prx1^{cre}Tet2^{fl/fl}$ mice, and the bone volume of $Prx1^{cre}Tet2^{fl/fl}$ mice was lower than that of control group (Fig. 2a-2c and Supplementary Fig.2a)."

To make the description more precise, we added the following sentences in the Introduction on page 3: "Additionally, 75% of the newborn *Tet1*-depleted mice exhibited a smaller body size at birth but seemed to gain their body weight after 6-9 weeks of life (Dawlaty and others, 2011), suggesting a potential developmental skeletal defect."

We also re-labeled "control mice" as " $Prx1^{cre}$ " in the Figure legend.

4. Another major concern is that the isolation of BM-MSc by just concentrating for the cells that stick to plastic would lead to a very heterogenous population-there is no FACS sorting or even just a single cell FACS based analyses to assess both the stemness and the heterogeneity of the populations. This is a major issue with all the in vitro experiments with the bone marrow derived MSCs. Sca1 and PDGFR have been utilized previously to isolate mouse MSC. See below-<http://www.nature.com/nprot/journal/v7/n12/full/nprot.2012.125.html?elq=b1a4061545a347c88a84111633b44f05>

Response: We appreciate the reviewer's suggestion. To verify the stem cell properties of isolated BMMSCs, we used flow cytometric analysis to show that they are negative for hematopoietic lineage markers CD34 and CD45 and positive for mesenchymal stem cell surface markers Sca1, PDGFR, CD105, CD90, and CD73. The newly generated data were added to the Results section

on page 6: “Flow cytometric analysis showed that BMMSCs both from control and *Tet* DKO mice were positive for stem cell surface markers Sca1, PDGFR, CD105, CD90, and CD73, but were negative for hematopoietic lineage markers CD34 and CD45 (Supplementary Fig. 2c)(Houlihan and others, 2012).”

5. The osteopenia phenotype after OVX in wild-type mice is very prominent compared to the TETDKO mouse-the authors should characterize OVX in the mutant mouse to differentiate potential developmental defects that the KO mice harbor from strictly postnatal MSC based bone regeneration.

Response: We appreciate the concerns raised by the reviewer. Because the *Tet* DKO mice were from the C57BL/6 strain whereas the OVX mice used in this study were C3H/HeJ, the femur bones of these two strains were different. To improve our ability to compare these different models, we conducted OVX on C57BL/6 wild-type and *Tet* DKO mice and reanalyzed the bone phenotype.

[Redacted]

We also added the following sentences to the Discussion section on page 14: “It was reported in a previous study that newborn *Tet1* and *Tet2* conventional knockout mice did not show developmental delay although a fraction of these double knockout mice were smaller in size (Dawlaty and others, 2013), suggesting that the osteopenia phenotype observed in *Tet* DKO mice

here may be attributed to the impairment of BMMSC osteogenic differentiation.

[Redacted]

6. Exosome characterization is very weak-FACS based analysis using CD9 and CD81 is required. The exosome isolation method also needs further details and some quality control to assess the enrichment of exosomes and the variation in preparation across samples. Present data with the western blot is of poor quality and needs quantitation and an independent technique to verify.

Response: We thank the reviewer for the constructive comments. We have added more details in the Methods section on page 18. Also, we used CD9 and CD81 intracellular immunostaining as well as an EXOCEP exosome quantitation kit (System Biociences Inc.) to quantitate the exosomes. We have added these newly generated data to the Results section on page 10: “Furthermore, we used an EXOCEP exosome quantitation kit to show that the exosome secretion was decreased in *Tet* DKO BMMSCs (Fig. 5c)(Witwer and others, 2013).” Also, we have re-conducted the Western blots and replaced Fig.5b and 5n accordingly.

Minor concerns

1. How are *Runx2*/*alp*/*ocn* regulated in the wildtype OVX model upon osteopenia induction-what are the 5mC/5hmC/C dynamics at their regulatory regions.

Response: We thank the reviewer for the thoughtful question. To investigate how *Runx2*/*ALP*/*OCN* are regulated in the WT OVX model upon osteopenia, we evaluated the expression levels of *Runx2*, *ALP*, and *OCN* in sham-treated and OVX BMMSCs and found decreased *Runx2* expression in OVX BMMSCs, but expression levels of *ALP* and *OCN* were not significantly different from those of the sham group. After osteogenic induction for 14 days, the expression levels of *Runx2*, *ALP*, and *OCN* were all downregulated. We next analyzed the *Runx2* promoter using Methprimer software and found that there was no CpG island. Also, we found through ChIP-qPCR that Tet1 and Tet2 fail to bind to the *Runx2* promoter, indicating that *Runx2* may be not the direct target of Tet. The following sentences were added to the Results section on page 8: “Next, to evaluate whether *Runx2* may be a direct target for Tet1 and Tet2, we analyzed

the *Runx2* promoter using Methprimer software and found that it lacks a CpG island (Supplementary Fig. 3e). We next used ChiP-qPCR to analyze whether Tet1/Tet2 could directly binding to the site where CpG is comparatively rich in the *Runx2* promoter. The result showed that no binding site was detected (Supplementary Fig. 3f). Thus, we searched for potential molecules that may connect Tet with Runx2(Kohli and Zhang, 2013; Xu and others, 2011).”

2. Loss of 5hmC in WT ovx (sup Fig 2) not convincing-a quantitative ELISA assay is needed.

Response: We appreciated the reviewer for this suggestion. To examine the reduced 5hmC level in the OVX group, we co-stained the 5hmC with stem cell surface marker CD146. We also used a dot blot assay to quantify the 5hmC levels in sham and OVX BMMSCs. We have added the newly generated data to the Results section on page 5: “We detected a reduced level of 5hmC in OVX mouse bone marrow stem cells using immunostaining and dot blot assay (Supplementary Fig. 1j-1k), consistent with the fact that Tet proteins convert 5-mC to 5-hmC.”

3. How was the brdU labeling done and how are the BMMSC populations isolated and analyzed for these assays? It is important to analyze whether the proliferation change is limited to/due to a subpopulation.

Response: We thank the reviewer for the thoughtful comment. We have added the detailed methods on page 21-22 accordingly.

We did not find significant differences among the groups of BMMSCs used in this study (Supplementary Fig. 2c). The proliferation change may reflect the bulk MSCs rather than being limited to a specific subpopulation.

4. Fig 4d- quantitation needed since the data does not convincing show an effect of the MiRNA inhibitors on Runx expression-why is the transcription data not collected/analyzed?

Response: The quantitation of the Western blot result was added to Supplementary Fig.3j, and the *Runx2* transcription data were added to Fig.4e and Fig. 4h.

5. Is the affect of miRNAs restricted to Runx since a change in alp/Ocn always accompanies Runx downregulation. Inhibitor treatment data on Day 0 and early time points of osteogenic differentiation is needed to assess a direct and specific effect on Runx 2.

Response: We appreciate the reviewer's thoughtful suggestion. To examine whether ALP and OCN are direct targets or downstream of Runx2, we compared the expression levels of *Runx2*, *ALP*, and *OCN* on different days after osteogenic induction. The newly generated data were added to the Results section on page 8: "After 14 days of osteogenic induction, the expression levels of *Runx2*, *ALP* and *OCN* were decreased in *Tet* DKO BMMSCs compared to the control group (Fig 3d). We compared the expression levels of *Runx2*, *ALP*, and *OCN* on different days after osteogenic induction to investigate whether *ALP* and *OCN* are potential direct targets or otherwise downstream of *Runx2*. The expression levels of *Runx2*, *ALP*, and *OCN* were markedly reduced after 14 days of osteogenic induction (Supplementary Fig. 3c), suggesting that *ALP* and *OCN* may be downstream targets of *Runx2*."

We also added the *Runx2*, *ALP*, and *OCN* expression data from control and miRNA inhibitor-treated BMMSCs after 3 days of osteogenic induction. We found a significantly decreased expression level of *Runx2*. We have added this newly generated data to Supplementary Fig. 3e.

Reviewer #3 (Remarks to the Author):

Yang et al report that inhibition of Tet1 and Tet2 results in impaired bone differentiation of bone marrow MSCs (BMMSCs) and a significant osteopenia phenotype. The authors further claim that Tet1 and Tet2 deficiency reduces demethylation of the P2rX7 promoter and thus downregulates exosome release, leading to intracellular accumulation of miR-297a-5p, miR-297b-5p, and miR-297c-5p and these miRNAs inhibit Runx2 signaling to impair BMMSC function. The manuscript seems to provide a novel understanding of how BMMSCs regulate normal skeletal homeostasis and should be of great interest to the readers; however, the authors need to provide further evidence to support their conclusion.

Response: We appreciate the reviewer's positive comments on the novelty of our work.

Major points:

1. The authors claim that Tet1 and Tet2 inhibition reduces demethylation of the promoter of P2rX7 that regulates exosome release, leading to intracellular accumulation of miR-297a-5p, miR-297b-5p, and miR-297c-5p. To support their conclusion, they need to demonstrate that

(a) exosomes released from control MSCs contain miR-297a-5p, miR-297b-5p, and miR-297c-5p and

Response: We appreciate the reviewer's concern. We compared the levels of miR-297a-5p, miR-297b-5p, and miR-297c-5p in exosomes and BMMSCs using qPCR and found that the levels of miR-297a-5p, miR-297b-5p, and miR-297c-5p in exosomes were higher than in BMMSCs (Attached Figure 3)

Attached Figure 3. qPCR analysis showed that the levels of miR-297a-5p, miR-297b-5p, and miR-297c-5p in exosomes were higher than in BMMSCs.

(b) treatment of these exosomes derived from control MSCs, but not the Tet DKO MSC-derived exosomes, suppresses osteogenic differentiation in control MSCs.

Response:

[Redacted]

Several studies have shown that differently enriched regulatory miRNAs from MSC-derived exosomes exert different biological effects on stem cell survival and differentiation (Davis et al. 2017; Liu et al. 2015; Qin et al. 2016). The different effects of a 50 µg/ml concentration of exosomes from control and *Tet* DKO MSCs on osteogenic differentiation imply that, to some extent, the selective intracellular accumulation of miR-297a-5p, miR-297b-5p, and miR-297c-5p in *Tet* DKO MSCs hampers the inhibitive effect of exosomes on osteogenic differentiation.

Parts of this peer review file have been redacted as indicated.

[Redacted]

(c) In addition, it would be important to show that Tet knockdown does not affect adipogenic differentiation potential in BMMSCs. Therefore, the authors need to show the result of adipogenic differentiation of either Tet siRNA transfected BMMSCs or Tet DKO mouse derived BMMSCs compared to their control BMMSCs.

Response: We appreciate the reviewer's suggestion. We conducted the adipogenic induction experiment using control and *Tet* DKO BMMSCs and added the data to the Results section on pages 6-7: "Control and *Tet* DKO BMMSCs showed similar adipogenic differentiation capacities under adipogenic induction, as indicated by Oil Red O staining and the expression of adipogenic-related genes lipoprotein (*LPL*) and peroxisome proliferator-activated receptor γ 2 (*PPAR* γ 2) (Supplementary Fig. 2d-2e)."

(d) Furthermore, I think it would be important to confirm these key observations with human BMMSCs as the authors mention that the Tet/P2rX7/Runx2 cascade may serve as a target for the development of novel therapies for osteopenia disorders.

Response: We appreciate the point raised by the reviewer. We used Tet1 and Tet2 siRNA together to treat human BMMSCs and found that the expression levels of *P2rX7* and *Runx2* were decreased, as shown in attached Figure 5.

Attached Figure 5. Western blot analysis showed that Tet1 and Tet2 combination siRNA treatment reduced the expression levels of *P2rX7* and *Runx2* in hBMMSCs.

The following sentences were added to the Discussion section on page 17: "The expression levels of *P2rX7* and *Runx2* were decreased in hBMMSCs after Tet1 and Tet2 combination siRNA treatment (data not shown)."

2. The authors claim that depletion of Tet1 and Tet2 impairs self-renewal and differentiation of BMSCs as they observed that Tet DKO mouse derived BMMSCs show an increase in CFU-f and BrdU positive cells in Figures 3a and 3b while showing the reduced bone differentiation. However, since Mahaira et al (Mahaira and others, 2014) previously reported the opposite result that inhibition of TET1/2 suppresses proliferation and Nanog levels in umbilical cord (UC)-derived MSCs, it would be important to provide strong evidence to support their conclusion. Previously, Moran-Crusio et al showed that Tet2 loss leads to increased hematopoietic stem cell self-renewal and myeloid transformation (Moran-Crusio and others, 2011). Hence, if the early passage BMMSC cultures contain the transformed hematopoietic cells, it could affect the results of CFU-f, BrdU and differentiation assays. Indeed, mouse MSCs isolated by the classical method of plastic adherence are frequently contaminated by overgrown hematopoietic cells(Phinney and others, 1999). Since the authors used passage one mouse MSCs for the current study, it would be important to demonstrate that the increased CFU-f and BrdU positive cells and decreased osteogenic differentiation in Tet DKO BMMSC culture are not due to contamination of the transformed hematopoietic cells. Therefore, MSC characterization data including cell morphology, adipogenic differentiation and flow cytometry analysis for surface epitopes of MSC markers (CD90, CD105 and CD73) and hematopoietic lineage markers (CD34 and CD45) need to be provided.

Response: We appreciate the reviewer's suggestion. To verify the stem cell properties of isolated BMMSCs, we used immunofluorescence staining with tubulin antibody to reveal the that morphology of BMMSCs in attached Figure 6.

Attached Figure 6. Immunofluorescence staining with tubulin antibody and DAPI showing that the morphology of control and *Tet* DKO BMMSCs. Scale bar: 20 μ m

Furthermore, we used flow cytometric analysis to show that hematopoietic lineage markers CD34 and CD45 were absent in BMMSCs whereas expression levels of stem cell surface markers Sca1, PDGFR, CD105, CD90, and CD73 were high. We have added these newly generated experimental data to the Results section on page 6 : “Flow cytometric analysis showed that BMMSCs both from control and *Tet* DKO mice were positive for stem cell surface markers Sca1, PDGFR, CD105, CD90, and CD73, but were negative for hematopoietic lineage markers CD34 and CD45 (Supplementary Fig. 2c)(Houlihan and others, 2012).” and “Control and *Tet* DKO BMMSCs showed similar adipogenic differentiation capacities under adipogenic induction, as indicated by Oil Red O staining and the expression of adipogenic-related genes lipoprotein (*LPL*) and peroxisome proliferator-activated receptor γ 2 (*PPAR* γ 2) (Supplementary Fig. 2d-2e).” The following sentences were also added to the Discussion section on page 14: “Previous studies showed that loss of *Tet2* leads to increased replicating capacity in hematopoietic stem cells and to myeloid transformation (Ko and others, 2010; Moran-Crusio and others, 2011). Inhibition of *Tet1/2* suppresses proliferation of umbilical cord (UC)-derived MSCs (Mahaira and others, 2014) and *Nanog* expression in embryonic stem cells (ES) (Ito and others, 2010). Here we showed that *Tet1* and *Tet2* depletion elevated the proliferation of BMMSCs. A previous study also reported that MSCs of adipose tissue, BM, and UC origin showed differential patterns of *IGF2BP1* expression (Mahaira and others, 2014), implying that the different effects of *Tet1/2* on stem cell proliferation may be regulated by *IGF2BP1* or other molecules.”

3. They further claim that *Tet* DKO mice show a significant osteopenia phenotype. However, *Tet* DKO seems to have a marginal effect on bone density in mice based on the results in Figure 2. To support their conclusion, the authors need to provide more convincing in vivo data. In addition, they need to show a proof of knockout of the target genes in transgenic mice.

Response: We appreciate the reviewer’s concern. We collected more samples and analyzed the bone mineral density (BMD), bone volume/tissue volume (BV/TV) (Fig. 2a), cortical bone area (Ct.Ar), cortical thickness (Ct.Th) (Fig. 2b) and distal femoral trabecular bone volume by microCT and histological analysis. The newly generated data were added to the Results section on page 5: “..., we compared the bone phenotypes of *Prx1*^{cre} (control), *Tet1*^{-/-}, *Prx1*^{cre}*Tet2*^{fl/fl}, and *Tet1*^{-/-};*Prx1*^{cre}*Tet2*^{fl/fl} double knockout (*Tet* DKO) mice at 8-10 weeks of age. Micro-CT and

histological analysis showed that *Tet* DKO, but not *Tet1*^{-/-}, mice had significantly decreased bone mineral density (BMD), bone volume/tissue volume (BV/TV), cortical bone area (Ct.Ar), cortical thickness (Ct.Th) and distal femoral trabecular bone volume compared to littermate controls (Fig. 2a-2c). *Tet* DKO mice showed significantly reduced BMD, BV/TV, Ct.Ar and distal femoral trabecular bone volume relative to *Tet1*^{-/-} and *Prx1*^{cre}*Tet2*^{fl/fl} mice; the bone volume of *Prx1*^{cre}*Tet2*^{fl/fl} mice was also less than that of controls (Fig. 2a-2c and Supplementary Fig.2a).”

Our newly generated data also showed that bone formation marker Runx2 decreased in the femurs of *Tet* DKO mice compared to controls. We have added following sentence to the Results section on page 8: “There were also fewer Runx2-positive cells in the femurs of *Tet* DKO mice compared to control ones (Supplementary Fig. 4d).”

Moreover, we added the mice genotyping and information about Tet1 and Tet2 depletion efficacy in BMMSCs in Supplementary Fig 2a-2b.

4. The authors claim that P2rX7 AAV increases secretion of exosomes in Tet DKO BMMSCs based on their western blot assay result in Figure 6h. However, the authors need to provide quantitative data of the western blotting results with statistical analysis since I don't see much difference in CD9 levels in Figure 6h.

Response: We appreciate the reviewer’s concern. We repeated the CD9 Western blot in Figure 6h, adding newly generated data and quantitative data in Supplementary Figure 5g.

Minor points:

The authors need to provide the following information;

- Sequence of primers for real-time PCR

Response: The primer sequences we used for qPCR, ChIP-PCR, miRNA, Ox-BS sequencing, RT-PCR and mouse genotyping were added to Supplementary Table 2.

- A cell source of human BMMSCs

Response: We provided the isolation and culture details for human BMMSCs in Materials and Methods section on page 19.

- A method of BrdU labeling assay

Response: We have added the detailed method for the BrdU labeling assay in pages 21-22.

- An explanation of how new bone formation was quantitated in Figure 3e and supplementary Figure 2e.

Response: We thank the reviewer for this valuable suggestion. We have added the details on page 25.

- Information of primary antibodies for immunofluorescence staining

Response: We have added information about the primary antibodies in the Materials and Methods section on page 18.

Reviewer #4 (Remarks to the Author):

The study by Shi and colleagues focuses on the role of Tet1 and Tet2 in BMMSC biology and its implications in osteopenia. They find that Tet1 and Tet2 are expressed in BMMSC and contribute to 5hmC levels. They show that loss of Tet1 and Tet2 in BMMSCs in mice leads to increased osteopenia. Mechanistically they show that Runx2 levels are down regulated in BMMSC due to increased levels of inhibitory microRNAs that accumulate in the cell because of impaired exosome release. They find silencing of P2rx7 gene, which is a mediator of exosome release, and establish that Tet1/2 is responsible for proper hydroxylation of P2rx7 promoter and its expression.

While the study is carefully carried out, it only focuses on one selected pathway. If the goal of the study is to define the role of Tet enzymes in BMMSC biology then more in depth and broader approaches should be considered. As it is, it does not comprehensively address the role of Tet enzymes in BMMSC biology as the authors propose to achieve in this study.

Main points.

1. What is the level of expression of Tet enzymes in BMMSC relative to other tissues or stem cell types such as embryonic or neural stem cells?

Response: We appreciate the reviewer's concern. We analyzed the expression levels of Tet1 and Tet2 in BMMSCs, neural stem cells and embryonic stem cells (ES) and showed that the expression levels of Tet1 and Tet2 were similar in BMMSCs and neural stem cells (NSC), but were lower than those in ES (Attached Figure 7).

Attached Figure 7. Western blot analysis showing the expression of Tet1 and Tet2 in neural stem cells, embryonic stem cells (ES) and BMMSCs.

2. What are the overall levels of 5hmC in BMMSCs and how much of it is contributed by Tet1 and Tet2? Immunofluorescence images are not very clear and do not provide quantitative data. Moreover, how does loss of Tet-mediated hydroxylation influence 5mC levels?

Response: We appreciate the points raised by the reviewer. We have added the newly generated data in the Result section on page 11 “Moreover, we detected the overall level of 5hmC and 5mC in BMMSCs by dot blot assay and showed that the level of 5hmC was lower in *Tet* DKO BMMSCs than the control group, while the level of 5mC was elevated in *Tet* DKO BMMSCs (Supplementary Fig. 4f-4g).” and the following sentences were added in the Discussion on page 16: “Tet1 and Tet2 deficiency also reduced the overall level of 5hmC in BMMSCs with downregulated expression of a variety of molecules, more investigation is required to illustrate the downstream target molecules of Tet1 and Tet2 in BMMSCs.”

3. How does loss of Tet1 and Tet2 influence the global gene expression programs in BMMSCs? Are there other mechanisms other P2rx7 and Runx2 involved? Some form of global gene expression analyses would provide better insight.

Response: We appreciate the points raised by the reviewer. We compared the global gene expression of control and Tet1 and Tet2 siRNA treated-BMMSCs by using RNA-sequencing. The newly generated data were added in the Results on pages 7-8: “To explore the underlying molecular mechanism, we performed RNA-sequencing analysis using RNA from control and Tet1/Tet2 siRNA-treated BMMSCs and found that around 80% of altered genes ($p < 0.05$ and fold change > 2) were decreased compared to the control group (Supplementary Fig. 3a). Functional analysis using WebGestalt showed that 19 of the 40 most significant enriched phenotype categories were related to skeletal bone/cartilage development. These altered genes, including *Runx2*, *ALP*, *Mmp2*, *Msx2*, *Sp7*, and *P2rx7*, are strongly associated with abnormal skeleton development. *Runx2* is one of the most significantly altered genes in all of the 19 phenotype categories (Supplementary Fig. 3b and Supplementary Table 3).”

The following sentences were added to the Discussion on page 17: “Global gene analysis showed *Runx2* cluster of genes were altered after Tet1 and Tet2 knockdown in BMMSCs, implies that the Tet/P2rX7/Runx2 cascade may be one of critical regulating mechanisms in maintaining bone homeostasis and BMMSC function.”

In addition, we added the experimental details to the Materials and Methods section on page 21 and deposited the RNA-sequencing data in the Gene Expression Omnibus (GEO) database.

4. How does changes in hydroxylation affect methylation in BMMSCs and how does that influence gene expression? I understand that the authors focus only on hydroxylation of P2rX7, but loss of these enzymes can influence many genes and loci globally with possible direct or indirect implications in BMMSC properties.

Response: We appreciate the reviewer’s concern. We have added the newly generated data that addresses this issue to the Results section on page 11: “Moreover, we detected the overall levels of 5hmC and 5mC in BMMSCs by dot blot assay and showed that the level of 5hmC was lower in *Tet* DKO BMMSCs than the control group, while the level of 5mC was elevated in *Tet* DKO BMMSCs (Supplementary Fig. 4f-4g).” The new data discussed in response to point 3 above also address this concern.

In addition, we added the experimental details to the Materials and Methods section on page 21 and deposited the RNA- sequencing data in the Gene Expression Omnibus (GEO) database.

5. The in vivo phenotype, and also the rescue results, though significant, are very marginal. The authors should comment on this as to how such small changes influence BMMSC and contribute to increased osteopenia.

Response: We appreciated the reviewer’s concern. We collected more samples and analyzed the bone mineral density (BMD), bone volume/tissue volume (BV/TV), cortical bone area (Ct.Ar), cortical thickness (Ct.Th) and distal femoral trabecular bone volume using MicroCT and histological analysis. These data have been added to Fig. 2.

We also added new data to the Results section on Page 8: “There were fewer *Runx2*-positive cells in the femurs of *Tet* DKO mice than in control ones (Supplementary Fig. 4d).”

The following sentence was added to the Discussion on page 14: “*Tet* DKO BMMSCs exhibited a significant impairment in osteogenic differentiation, which contributed to the osteopenia phenotype observed on *Tet* DKO mice.”

6. If the mechanism is through hydroxylation of P2rX7 promoter the authors should show if it is the hydroxylation or the subsequent demethylation that allows for proper expression of P2rX7. I think 5mC levels at promoter should be quantified too. Also locus specific sequencing for 5hmC and 5mC should be done instead of DIP-PCR.

Response: We thank the reviewer for the suggestion. We analyzed the P2rX7 after Tet1 and Tet2 overexpression and the data were added on the Result section on Page 11: “Furthermore, we overexpressed wildtype Tet1 and Tet2 plasmid and catalytic domain inactive Tet1 and Tet2 plasmids in *Tet* DKO BMMSCs to analyze whether overexpression could rescue the decreased expression of P2rX7 and osteogenic differentiation. The results showed that overexpression of wildtype Tet1 and Tet2 plasmids, but not catalytic domain inactive ones, overexpression rescued decreased expression of P2rX7 in *Tet* DKO BMMSCs (Supplementary Fig. 4h).”

The ChiP-qPCR analysis for 5mC and OxBS-sequencing analysis of the methylation status of *P2rX7* promoter were added to the Results section on page 11: “MeDIP-qPCR analysis revealed that *Tet* DKO BMMSCs showed increased 5mC levels compared to control BMMSCs (Supplementary Fig. 4d). Ox-BS sequencing analysis also showed that *Tet* DKO BMMSCs displayed elevated methylation in the promoter of *P2rx7* locus compared to control BMMSCs (Supplementary Fig. 4e).”

7. I do not see a rescue of phenotypes by Tet1 or Tet2 re-expression. This would be useful, especially if catalytic mutant versions of Tet enzymes are tested too.

Response: We appreciate the concern raised by the reviewer. We analyzed the osteogenic differentiation capacity of *Tet* DKO BMMSCs after Tet1 and Tet2 overexpression and the data were added on the Result section on Page 11: “Furthermore, we overexpressed wildtype Tet1 and Tet2 plasmid and catalytic domain inactive Tet1 and Tet2 plasmids in *Tet* DKO BMMSCs to analyze whether overexpression could rescue the decreased expression of P2rX7 and osteogenic

differentiation. The results showed that overexpression of wildtype Tet1 and Tet2 plasmids, but not catalytic domain inactive ones, overexpression rescued decreased expression of *P2rx7* in *Tet* DKO BMMSCs (Supplementary Fig. 4h). Wildtype, but not catalytic domain inactive, Tet1 and Tet2 plasmid overexpression also elevated the mineralized nodule formation and expression levels of *Runx2*, *ALP*, and *OCN* under osteogenic induction (Supplementary Fig. 4h-4j).”

8. Are the levels of miRNAs increased because of exosome release issue only? Or that Tet enzymes can regulate their levels directly as the authors mention in the discussion.

Response: We appreciate the points raised by the reviewer. Song et al. have reported that Tet directly regulates miR-200 transcription by controlling the 5hmC levels (Song and others, 2013). We looked for but did not detect CpG islands in the promoters of miR-297a-5p, miR-297b-5p, and miR-297c-5p using methprimer analysis by the Li lab. The following sentences were added to the Results on page 9: “We could not detect a CpG island on the promoter of these three microRNAs, indicating that they may not be directly targeted by Tet1 or Tet2.” The following sentence was added to the Discussion on page 16: “In this study, we show that there are no CpG islands in the promoters of miR-297a-5p, miR-297b-5p, or miR-297c-5p, and we reveal a new mechanism of interaction between the Tet family and miRNAs through which Tet1 and Tet2 depletion lead to *P2rx7* promoter hypermethylation and reduced exosome secretion.”

Other points:

1. I do not find clear images showing that the impaired release of exosome leads to increased number of exosomes inside the cells. They only show less exosomes released.

Response: We appreciate the points raised by the reviewer. In order to analyze whether the exosomes accumulate in *Tet* DKO BMMSCs, we generated new data using immune-fluorescence staining and transmission electron microscopy. The results were added to the Results section on page 10: “More intracellular exosomes accumulated in *Tet* DKO BMMSCs compared to control BMMSCs, as indicated by more CD9-positive and CD81-positive intracellular exosomes of *Tet*

DKO BMMSCs, as analyzed by immunofluorescence staining, and more vesicles accumulation, as analyzed by transmission electron microscopy (Fig. 5d-5e).”

2. The deletion efficiency of Tet2 floxed allele is not shown. Also the ages at which the analyses are done are not clearly documented.

Response: We thank for the reviewer for this attention to detail. The deletion efficiency of Tet1 and Tet2 based on mice genotyping and Western blot were added to Supplementary Figure 2a-2b. Detailed information about the ages of mice in each experiment was added to each appropriate Figure legend.

3. The cartoon depicting the model is very misleading. There should be side-by-side models for presence and absence of Tet1/2 to clearly illustrate the story.

Response: We appreciate the reviewer’s suggestion and revised the cartoon accordingly.

References:

- Agrawal A, Henriksen Z, Syberg S, Petersen S, Aslan D, Solgaard M, Nissen N, Larsen TK, Schwarz P, Steinberg TH and others. 2017. P2X7Rs are involved in cell death, growth and cellular signaling in primary human osteoblasts. *Bone* 95:91-101.
- An J, Rao A, Ko M. 2017. TET family dioxygenases and DNA demethylation in stem cells and cancers. *Exp Mol Med* 49(4):e323.
- Dawlaty MM, Breiling A, Le T, Raddatz G, Barrasa MI, Cheng AW, Gao Q, Powell BE, Li Z, Xu M and others. 2013. Combined deficiency of Tet1 and Tet2 causes epigenetic abnormalities but is compatible with postnatal development. *Dev Cell* 24(3):310-323.
- Dawlaty MM, Ganz K, Powell BE, Hu YC, Markoulaki S, Cheng AW, Gao Q, Kim J, Choi SW, Page DC and others. 2011. Tet1 is dispensable for maintaining pluripotency and its loss is compatible with embryonic and postnatal development. *Cell Stem Cell* 9(2):166-175.
- Delhommeau F, Dupont S, Della Valle V, James C, Trannoy S, Masse A, Kosmider O, Le Couedic JP, Robert F, Alberdi A and others. 2009. Mutation in TET2 in myeloid cancers. *N Engl J Med* 360(22):2289-2301.
- Houlihan DD, Mabuchi Y, Morikawa S, Niibe K, Araki D, Suzuki S, Okano H, Matsuzaki Y. 2012. Isolation of mouse mesenchymal stem cells on the basis of expression of Sca-1 and PDGFR-alpha. *Nat Protoc* 7(12):2103-2111.
- Ito S, D'Alessio AC, Taranova OV, Hong K, Sowers LC, Zhang Y. 2010. Role of Tet proteins in 5mC to 5hmC conversion, ES-cell self-renewal and inner cell mass specification. *Nature* 466(7310):1129-1133.
- Ko M, Huang Y, Jankowska AM, Pape UJ, Tahiliani M, Bandukwala HS, An J, Lamperti ED, Koh KP, Ganetzky R and others. 2010. Impaired hydroxylation of 5-methylcytosine in myeloid cancers with mutant TET2. *Nature* 468(7325):839-843.
- Kohli RM, Zhang Y. 2013. TET enzymes, TDG and the dynamics of DNA demethylation. *Nature* 502(7472):472-479.
- Lenertz LY, Gavala ML, Zhu Y, Bertics PJ. 2011. Transcriptional control mechanisms associated with the nucleotide receptor P2X7, a critical regulator of immunologic, osteogenic, and neurologic functions. *Immunol Res* 50(1):22-38.
- Li W, Li G, Zhang Y, Wei S, Song M, Wang W, Yuan X, Wu H, Yang Y. 2015. Role of P2 x 7 receptor in the differentiation of bone marrow stromal cells into osteoblasts and adipocytes. *Exp Cell Res* 339(2):367-379.
- Liu S, Liu D, Chen C, Hamamura K, Moshaverinia A, Yang R, Liu Y, Jin Y, Shi S. 2015. MSC Transplantation Improves Osteopenia via Epigenetic Regulation of Notch Signaling in Lupus. *Cell Metab* 22(4):606-618.
- Mahaira LG, Katsara O, Pappou E, Iliopoulou EG, Fortis S, Antsaklis A, Fotinopoulos P, Baxevanis CN, Papamichail M, Perez SA. 2014. IGF2BP1 expression in human mesenchymal stem cells significantly affects their proliferation and is under the epigenetic control of TET1/2 demethylases. *Stem Cells Dev* 23(20):2501-2512.
- Moran-Crusio K, Reavie L, Shih A, Abdel-Wahab O, Ndiaye-Lobry D, Lobry C, Figueroa ME, Vasanthakumar A, Patel J, Zhao X and others. 2011. Tet2 loss leads to increased hematopoietic stem cell self-renewal and myeloid transformation. *Cancer Cell* 20(1):11-24.

- Phinney DG, Kopen G, Isaacson RL, Prockop DJ. 1999. Plastic adherent stromal cells from the bone marrow of commonly used strains of inbred mice: variations in yield, growth, and differentiation. *J Cell Biochem* 72(4):570-585.
- Song SJ, Poliseno L, Song MS, Ala U, Webster K, Ng C, Beringer G, Brikbak NJ, Yuan X, Cantley LC and others. 2013. MicroRNA-antagonism regulates breast cancer stemness and metastasis via TET-family-dependent chromatin remodeling. *Cell* 154(2):311-324.
- Tefferi A, Lim KH, Abdel-Wahab O, Lasho TL, Patel J, Patnaik MM, Hanson CA, Pardanani A, Gilliland DG, Levine RL. 2009. Detection of mutant TET2 in myeloid malignancies other than myeloproliferative neoplasms: CMML, MDS, MDS/MPN and AML. *Leukemia* 23(7):1343-1345.
- Wang L, Zhao Y, Liu Y, Akiyama K, Chen C, Qu C, Jin Y, Shi S. 2013. IFN-gamma and TNF-alpha synergistically induce mesenchymal stem cell impairment and tumorigenesis via NFkappaB signaling. *Stem Cells* 31(7):1383-1395.
- Witwer KW, Buzas EI, Bemis LT, Bora A, Lasser C, Lotvall J, Nolte-'t Hoen EN, Piper MG, Sivaraman S, Skog J and others. 2013. Standardization of sample collection, isolation and analysis methods in extracellular vesicle research. *J Extracell Vesicles* 2.
- Xu Y, Wu F, Tan L, Kong L, Xiong L, Deng J, Barbera AJ, Zheng L, Zhang H, Huang S and others. 2011. Genome-wide regulation of 5hmC, 5mC, and gene expression by Tet1 hydroxylase in mouse embryonic stem cells. *Mol Cell* 42(4):451-464.

Reviewer #4 (Remarks to the Author):

The authors have made a genuine effort to address the points that I had raised. While they do not completely address all the points, their revisions provide additional information to support their conclusions. I have summarized my comments below.

Major points

1- Addressed

2- Addressed

3- Addressed

4- Not completely addressed as the only way to fully show the correlation between gene expression and methylation genome wide is to examine global distribution of 5mC (possibly 5hmC) in control and mutant cells. They only quantify global levels (not global distribution or identify target genes)

Response: We appreciate the reviewer's suggestion to examine the global distribution of 5mC or 5hmC. Previous genome-wide 5hmC mapping in pluripotent stem cells and differentiated cells showed that Tet regulates 5hmC levels in the promoters and exons of targeted genes *via* regulating both gene transcriptional activation and repression in a context-dependent manner¹⁻⁶. The landscape sequencing analysis may provide abundant evidence for further PCR or Chip-qPCR analysis to validate the findings from wholegenome sequencing study². We agree with the reviewer's previous comment "**I understand that the authors focus only on hydroxylation of P2rX7, but loss of these enzymes can influence many genes and loci globally with possible direct or indirect implications in BMMSC properties.**" Therefore, we focused on hydroxylation of P2rX7 and explored the potential targets involved in the osteopenia phenotype. We used Ox-BS-seq and Chip-seq analysis to demonstrate that control and Tet mutant cells showed different enrichment and distribution of 5hmC and 5mC on the promoter of target genes. We further used different strategies to confirm this cascade is required to maintain bone homeostasis. Conducting genome wide sequencing would be costly and may not provide additional information to confirm that P2rX7 is a target gene.

We have added the following sentences in the Discussion section to explain other possible direct or indirect implications of the hydroxylation of genes: "As 5hmC may be involved in establishing and maintaining chromatin structure for both actively transcribed genes and PcG-repressed regulators, it also affects both transcriptional activation and repression roles in a context-dependent manner. The detailed mechanisms of how 5hmC regulates other genes altered by Tet in BMMSCs may need further investigation⁶⁻⁸."

5- In part addressed. I was more specifically asking how such a mild/marginal phenotype could cause osteopenia but authors have gone into a different direction.

Response: We appreciate the reviewer's suggestion. BMMSC impairment is sufficient to cause osteopenia phenotype. The following sentences were added to the Discussion

section on page 14:”suggesting that the osteopenia phenotype observed in Tet DKO mice may be attributed to the impairment of BMMSC osteogenic differentiation. Bone homeostasis is a tightly regulated process, balancing new bone formation by BMMSC-derived osteoblasts with bone resorption by osteoclasts. Signals that determine the recruitment, replication, apoptosis and differentiation of cells of both lineages may increase osteopenia risk^{9,10}. For instance, alteration of histone H3K9 acetyltransferase PCAF¹¹, mTOR signaling¹² and hydrogen sulfide levels¹³ may lead to osteopenia by impairing osteogenic differentiation of BMMSCs. “

6- In figure S4e: Normally Ox-BS-seq is used to measure the level of hydroxymethylation. I don't know why the authors indicate methylation in the legend and make no reference to 5hmC.

Response: The figure legend was revised accordingly.

7- This rescue experiment is not done with the right reference control. A WT cell line plus empty vector (vehicle) should have been used as reference control. Now we see that Tet1/2 increases the expression of Runx2 and improves the phenotype, but to what extent compared to WT is not known.

Response: We appreciate the reviewer's suggestion. The WT control was added accordingly in Supplementary Figure 4h-4j.

8- Addressed

Minor points:

1- Addressed.

2- Figure 2a is not properly annotated. How many clones or cell lines had full deletion of Tets? It appears it is only one clone.

Response: We appreciate the reviewer's suggestion. Supplementary Figure 2a is a representative of genotyping results for mice from one breeding cage. Around twenty Tet DKO mice were analyzed using different assays. [REDACTED] [REDACTED] [REDACTED] [REDACTED] [REDACTED] Figure legend for Supplementary Figure 2 was revised accordingly.

Editorial Note: Parts of this peer review file have been redacted as indicated.

[Redacted]

3- Addressed.

References

- 1 Wu, H., D'Alessio, A. C., Ito, S., Wang, Z., Cui, K., Zhao, K. *et al.* Genome-wide analysis of 5-hydroxymethylcytosine distribution reveals its dual function in transcriptional regulation in mouse embryonic stem cells. *Genes Dev* 2011; **25**: 679-684
- 2 Xu, Y., Wu, F., Tan, L., Kong, L., Xiong, L., Deng, J. *et al.* Genome-wide regulation of 5hmC, 5mC, and gene expression by Tet1 hydroxylase in mouse embryonic stem cells. *Mol Cell* 2011; **42**: 451-464
- 3 Pastor, W. A., Pape, U. J., Huang, Y., Henderson, H. R., Lister, R., Ko, M. *et al.* Genome-wide mapping of 5-hydroxymethylcytosine in embryonic stem cells. *Nature* 2011; **473**: 394-397
- 4 Choi, I., Kim, R., Lim, H. W., Kaestner, K. H. & Won, K. J. 5-hydroxymethylcytosine represses the activity of enhancers in embryonic stem cells: a new epigenetic signature for gene regulation. *BMC genomics* 2014; **15**: 670
- 5 Booth, M. J., Branco, M. R., Ficuz, G., Oxley, D., Krueger, F., Reik, W. *et al.* Quantitative sequencing of 5-methylcytosine and 5-hydroxymethylcytosine at single-base resolution. *Science* 2012; **336**: 934-937
- 6 Mellen, M., Ayata, P., Dewell, S., Kriaucionis, S. & Heintz, N. MeCP2 binds to 5hmC enriched within active genes and accessible chromatin in the nervous system. *Cell* 2012; **151**: 1417-1430
- 7 Wu, H. & Zhang, Y. Mechanisms and functions of Tet protein-mediated 5-methylcytosine oxidation. *Genes Dev* 2011; **25**: 2436-2452
- 8 Wu, H., D'Alessio, A. C., Ito, S., Xia, K., Wang, Z., Cui, K. *et al.* Dual functions of Tet1 in transcriptional regulation in mouse embryonic stem cells. *Nature* 2011; **473**: 389-393
- 9 Canalis, E. The fate of circulating osteoblasts. *The New England journal of medicine* 2005; **352**: 2014-2016

- 10 Crane, J. L. & Cao, X. Function of matrix IGF-1 in coupling bone resorption and formation. *J Mol Med (Berl)* 2014; **92**: 107-115
- 11 Zhang, P., Liu, Y., Jin, C., Zhang, M., Lv, L., Zhang, X. *et al.* Histone H3K9 Acetyltransferase PCAF Is Essential for Osteogenic Differentiation Through Bone Morphogenetic Protein Signaling and May Be Involved in Osteoporosis. *Stem cells* 2016; **34**: 2332-2341
- 12 Chen, C., Akiyama, K., Wang, D., Xu, X., Li, B., Moshaverinia, A. *et al.* mTOR inhibition rescues osteopenia in mice with systemic sclerosis. *J Exp Med* 2015; **212**: 73-91
- 13 Liu, Y., Yang, R., Liu, X., Zhou, Y., Qu, C., Kikuri, T. *et al.* Hydrogen Sulfide Maintains Mesenchymal Stem Cell Function and Bone Homeostasis via Regulation of Ca Channel Sulfhydration. *Cell Stem Cell* 2014;

REVIEWERS' COMMENTS:

Reviewer #4 (Remarks to the Author):

My points are addressed by the additional experiments and editing of figure legends and expansion of discussion.

The addition of wild type control in the rescue experiments shows that the degree of rescue is not full and is partial, specially for OCN and Runxl. Possibly other parallel mechanisms may contribute too as acknowledged in the discussion.

REVIEWERS' COMMENTS:

Reviewer #4 (Remarks to the Author):

My points are addressed by the additional experiments and editing of figure legends and expansion of discussion.

The addition of wild type control in the rescue experiments shows that the degree of rescue is not full and is partial, specially for OCN and Runxl. Possibly other parallel mechanisms may contribute too as acknowledged in the discussion.

Response: We appreciate the reviewer's comment. The following sentences were acknowledged in the Discussion section: "Collectively, our studies reveal that *P2rx7* is a target of Tet demethylation, and that Tet1 and Tet2 can directly modulate *P2rx7*, thereby controlling BMMSC exosome release to maintain bone and BMMSC homeostasis. Tet1 and Tet2 deficiency also reduced the overall level of 5hmC in BMMSCs with downregulated expression of a variety molecular, more investigation is required to illustrate other downstream target molecules of Tet1 and Tet2 in BMMSCs."